# Dinomaly: The *Less Is More* Philosophy in Multi-Class Unsupervised Anomaly Detection

## Abstract

Recent studies highlighted a practical setting of unsupervised anomaly detection (UAD) that builds a unified model for multi-class images, serving as an alternative to the conventional one-class-one-model setup. Despite various advancements addressing this challenging task, the detection performance under the multi-class setting still lags far behind state-of-the-art class-separated models. Our research aims to bridge this substantial performance gap. In this paper, we introduce a minimalistic reconstruction-based anomaly detection framework, namely Dinomaly, which leverages pure Transformer architectures without relying on complex designs, additional modules, or specialized tricks. Given this powerful framework consisted of only Attentions and MLPs, we found four simple components that are essential to multi-class anomaly detection: (1) *Foundation Transformers* that extracts universal and discriminative features, (2) *Noisy Bottleneck* where pre-existing Dropouts do all the noise injection tricks, (3) *Linear Attention* that naturally cannot focus, and (4) *Loose Reconstruction* that does not force layer-to-layer and point-by-point reconstruction. Extensive experiments are conducted across three popular anomaly detection benchmarks including MVTec-AD, VisA, and the recently released Real-IAD. Our proposed Dinomaly achieves impressive image AUROC of 99.6%, 98.7%, and 89.3% on the three datasets respectively, which is not only superior to state-of-the-art multi-class UAD methods, but also surpasses the most advanced class-separated UAD records.

## 1 Introduction

Unsupervised anomaly detection (UAD) aims to detect abnormal patterns from normal images and further localize the anomalous regions. Because of the diversity of potential anomalies and their scarcity, this task is proposed to model the accessible training sets containing only normal samples as an unsupervised paradigm. UAD has a wide range of applications, e.g., industrial defect detection, medical disease screening, and video surveillance, addressing the difficulty of collecting and labeling all possible anomalies in these scenarios.

Efforts on UAD attempt to learn the distribution of available normal samples. Most advanced methods utilize networks pre-trained on large-scale datasets, e.g. ImageNet [1], for extracting discriminative and informative feature representations. Specifically, *Feature reconstruction* [2; 3; 4] and *feature distillation* methods [5; 6] are proposed to reconstruct features of pre-trained encoders, based on the hypothesis that the networks trained on normal images can only construct normal regions, but fail for unseen anomalous regions. *Feature statistics* methods [7; 8; 9] memorize and model all anomaly-free features extracted from pre-trained networks in training, and compare them with the test features during inference. *Pseudo-anomaly* methods [10; 11] generate pseudo defects or noises

Submitted to 38th Conference on Neural Information Processing Systems (NeurIPS 2024). Do not distribute.

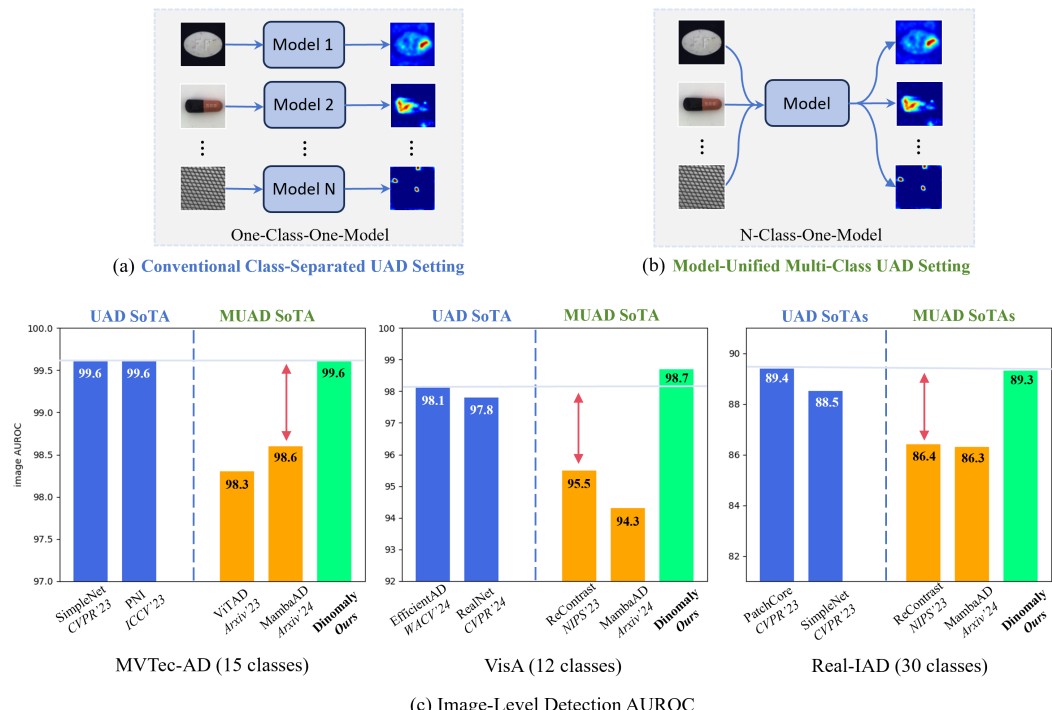

Figure 1: Setting and Performance of UAD and multi-class UAD (MUAD). (a) Task setting of class-separated UAD. (b) Task setting of MUAD. (c) Comparison of Dinomaly and previous SoTA methods [13; 14; 15; 16; 8; 17; 18; 19] on MVTec-AD [20], VisA [21], and Real-IAD [22].

on normal images or features to imitate anomalies, converting UAD to supervised classification [11] or segmentation tasks [10; 12].

Conventional works on UAD build a separate model for each object category, as shown in Figure 1(a). However, this one-class-one-model setting entails substantial storage overhead for saving models [3], especially when the application scenario necessitates a large number of object classes. For UAD methods, a compact boundary of normal patterns is vital to distinguish anomalies. Once the intra-normal patterns become exceedingly complicated due to various classes, the corresponding distribution becomes challenging to measure, consequently harming the detection performance. Recently, UniAD [3] and successive studies have been proposed to train a unified model for multi-class anomaly detection (MUAD), as shown in Figure 1(b). Under this setting, the "identity mapping" that directly copies the input as the output regardless of normal or anomaly harms the performance of conventional methods [3]. This phenomenon is caused by the diversity of multi-class normal patterns that drive the network to generalize on unseen patterns.

Within two years, a number of methods have been proposed to address MUAD, such as neighbor-masked attention [3], synthetic anomalies [23], feature jitter [3], vector quantization [24], diffusion model [25; 26], and state space model (Mamba) [19]. However, there is still a non-negligible performance gap between the state-of-the-art (SoTA) MUAD methods and class-separated UAD methods, restricting the practicability of implementing unified models, as shown in Figure 1(c). In addition, previous methods employ modules and architectures delicately designed, which may not be straightforward, and consequently suffer from limited universality and usability.

In this work, we aim to catch up with the performance of class-separated anomaly detection models using a multi-class unified model, namely Dinomaly. To begin with, we build a reconstruction-based UAD framework that consists of only vanilla Transformer blocks [27], i.e. Self-Attentions and Multi-Layer Perceptrons (MLPs). Within this framework, we propose four simple but essential elements that boost Dinomaly to perform equal to or better than SoTA conventional class-separated models. First, we show that self-supervised pre-trained Vision Transformers (ViT) [28], especially the DINO family [29; 30], serve as powerful feature encoders to extract discriminative representations as

reconstruction objects. Second, as an alternative to carefully designed pseudo anomaly and feature noise, we propose to activate the out-of-the-box Dropout in an MLP to prevent the network from restoring both normal and anomalous patterns, which is previously referred to as identity mapping. Third, we propose to utilize the "side effect" of Linear Attention (a computation-efficient counterpart of Softmax Attention) that makes it hard to focus on local regions, to further alleviate the issue of identity mapping. Fourth, previous methods adopt layer-to-layer and region-by-region reconstruction schemes, distilling a decoder that can well mimic the behavior of the encoder even for anomalous regions. Therefore, we propose to loosen the reconstruction constraints by grouping multiple layers as a whole and discarding well-reconstructed regions during optimization.

To validate the effectiveness of the proposed Dinomaly under MUAD setting, we conduct extensive experiments on three widely used industrial defect detection benchmarks, i.e., MVTec AD [20] (15 classes), VisA [21] (12 classes), and recently released Real-IAD (30 classes). Notably, we achieve unprecedented image-level AUROC of 99.6%, 98.7%, and 89.3% on MVTec AD, VisA, and Real-IAD, respectively, which surpasses previous SoTA methods by a large margin.

Related works are presented in Appendix A.1.

## 2 Method

### 2.1 Dinomaly Framework

*"What I cannot create, I do not understand"*——Richer Feynman

The ability to recognize anomalies from what we know is an innate human capability, serving as a vital pathway for us to explore the world. Similarly, we construct a reconstruction-based framework that relies on the epistemic characteristic of artificial neural networks. Dinomaly consists of an encoder, a bottleneck, and a reconstruction decoder, as shown in Figure 2. Without loss of generality, a standard ViT-Base/14 network [28] with 12 Transformer layers is used as the encoder, extracting informative feature maps with different semantic scales. The bottleneck is a simple MLP (a.k.a. feed-forward network, FFN) that collects the feature representations of the encoder's 8 middle-level layers. The decoder is similar to the encoder, consisting of 8 Transformer layers. During training, the decoder learns to reconstruct the middle-level features of the encoder by maximizing the cosine similarity between feature maps. During inference, the decoder is expected to reconstruct normal regions of feature maps but fails for anomalous regions as it has never seen such samples.

**Foundation Transformers.** Foundation models, especially ViTs [28; 31] pre-trained on large-scale datasets, serve as a basis and starting point for specific computer vision tasks. Such networks employ self-supervised learning schemes such as contrastive learning (MoCov3 [32], DINO [29]), masked image modeling (MAE [33], SimMIM [34], BEiT [35]), and their combination (iBOT [36], DINOv2 [30]), producing universal features suitable for image-level visual tasks (image classification, instance retrieval) and pixel-level visual tasks (depth estimation, semantic segmentation).

Because of the lack of supervision in UAD, most advanced methods adopt pre-trained networks to extract discriminative features. Recent works [37; 17; 38] have discovered the advantage of robust and universal features of self-supervised models over domain-specific ImageNet features in anomaly detection tasks. In this work, we further utilize the up-to-date Transformer foundation, i.e., DINOv2 with registers [39], as the encoder of Dinomaly.

### 2.2 Noisy Bottleneck.

*"Dropout is all you need."*

Generalization ability is a merit of neural networks, allowing them to perform equally well on unseen test sets. However, generalization is not so wanted in the context of unsupervised anomaly detection that leverages the epistemic nature of neural networks. With the increasing diversity of images and their patterns due to multi-class UAD settings, the decoder can generalize its reconstruction ability to unseen anomalous samples, resulting in the failure of anomaly detection using reconstruction error. This phenomenon is called "identity mapping" in previous works of literature [3; 23; 18].

A direct solution for identity mapping is to shift "reconstruction" to "restoration". Specifically, instead of directly reconstructing the normal images or features given normal inputs, previous works propose

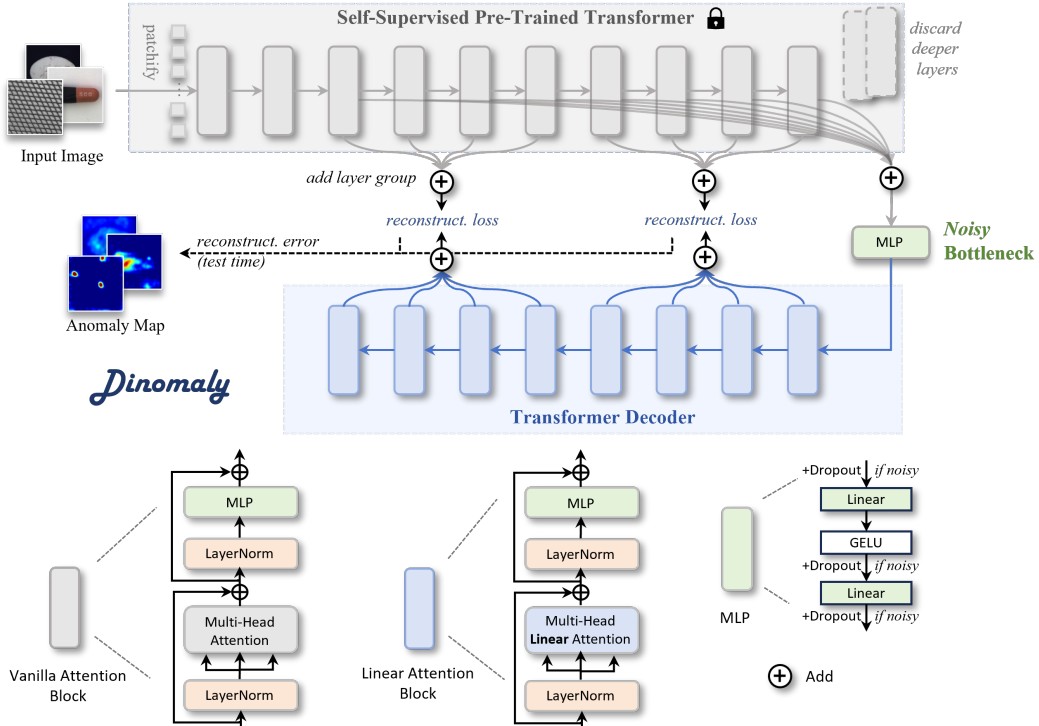

Figure 2: The framework of Dinomaly, built by pure Transformer building blocks.

to add perturbations as pseudo anomalies on input images [10; 40; 12] or feature representations [3; 25] during network forward propagation; meanwhile, still let the decoder restore anomaly-free images or features, formulating a denoising-like framework. However, such methods employ heuristic and hand-crafted anomaly generation strategies, that are not universal across domains, datasets, and methods.

In this work, we propose to activate the pre-existing Dropout in an MLP layer. Dropout, a popular network element introduced by Hinton et al. [41] in 2014 to prevent overfitting, flourished in nearly all neural network architectures to the present day, including Transformers. In Dinomaly, Dropout is used to discard neural activations in the MLP bottleneck randomly. Instead of alleviating overfitting, the role of Dropout in Dinomaly can be explained as feature noise and pseudo feature anomaly. Although the decoder takes noisy features during training, it is encouraged to restore clean features from the encoder. Without introducing any novel modules, this paradigm forces the decoder to restore normal features given a test image with anomalies, in turn, mitigating identical mapping.

## 2.3 Unfocused Linear Attention.

*"One man's poison is another man's meat"*

**Softmax Attention** is the key mechanism of Transformers, allowing the model to attend to different parts of its input token sequence. Formally, given an input sequence $\mathbf{X} \in \mathbb{R}^{N \times d}$ with length $N$, Attention first transforms it into three matrices: the query matrix $\mathbf{Q} \in \mathbb{R}^{N \times d}$, the key matrix $\mathbf{K} \in \mathbb{R}^{N \times d}$, and the value matrix $\mathbf{V} \in \mathbb{R}^{N \times d}$:

$$\mathbf{Q} = \mathbf{X}\mathbf{W}^Q , \mathbf{K} = \mathbf{X}\mathbf{W}^K , \mathbf{V} = \mathbf{X}\mathbf{W}^V , \tag{1}$$

where $\mathbf{W}^Q, \mathbf{W}^K, \mathbf{W}^V \in \mathbb{R}^{d \times d}$ are learnable parameters. By computing the attention map by the query-key similarity, the output of Attention is given as: [1]

$$\text{Attention}(\mathbf{Q}, \mathbf{K}, \mathbf{V}) = \text{Softmax}(\mathbf{Q}\mathbf{K}^T)\mathbf{V} . \tag{2}$$

---

[1]The full form of Attention is $\text{Softmax}(\frac{\mathbf{Q}\mathbf{K}^T}{\sqrt{d}})\mathbf{V}$. The constant denominator is omitted for narrative simplicity. The multi-head mechanism that concatenates multiple Attentions is also omitted.

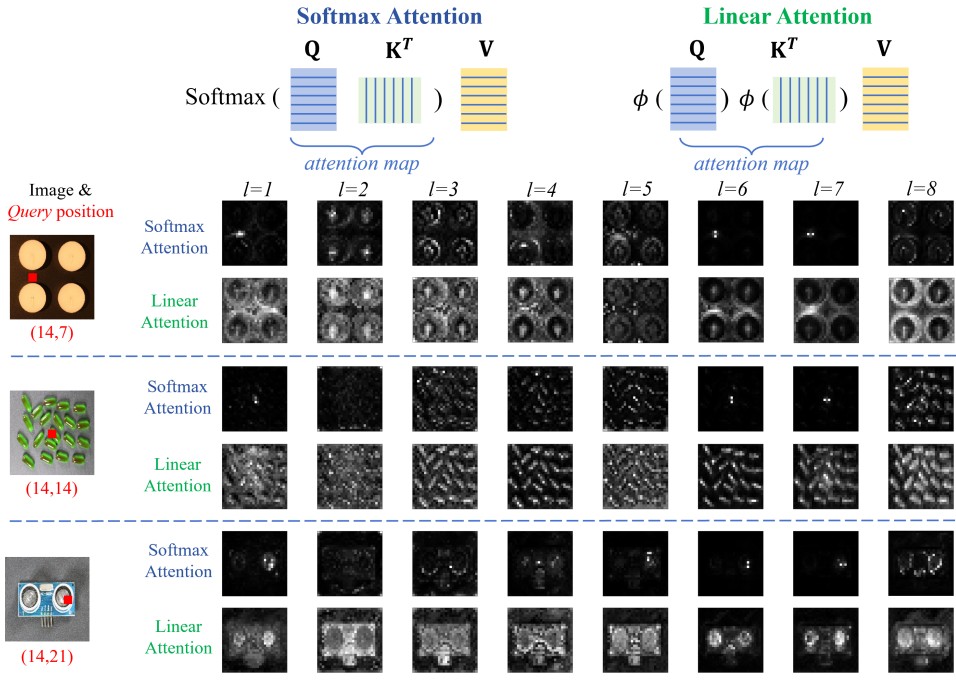

Figure 3: The decoder attention map (min-max to 0-1 for visualization) of Dinomaly with vanilla Softmax Attention *vs.* Linear Attention.

Because the attention map is obtained by computing the similarity between all query-key pairs followed by row-wise Softmax, the computation complexity is $\mathcal{O}(N^2 d)$.

**Linear Attention** was proposed as a promising alternative to reduce the computation complexity of vanilla Softmax Attention concerning the number of tokens [42]. By substituting Softmax operation with a simple activation function $\phi(\cdot)$ (usually $\phi(x) = \text{elu}(x) + 1$), we can change the computation order from $(\mathbf{Q}\mathbf{K}^T)\mathbf{V}$ to $\mathbf{Q}(\mathbf{K}^T\mathbf{V})$. Formally, Linear Attention is given as:

$$\text{LinearAttention}(\mathbf{Q}, \mathbf{K}, \mathbf{V}) = (\phi(\mathbf{Q})\phi(\mathbf{K}^T))\mathbf{V} = \phi(\mathbf{Q})(\phi(\mathbf{K}^T)\mathbf{V}) , \qquad (3)$$

where the computation complexity is reduced to $\mathcal{O}(Nd^2)$. The trade-off between complexity and expressiveness is a dilemma. Previous studies [43; 44] attribute Linear Attention's performance degradation on supervised tasks to its incompetence in focusing. Due to the absence of non-linear attention reweighting by Softmax operation, Linear Attention cannot concentrate on important regions related to the query, such as foreground and neighbors.

Back to MUAD, previous methods [3; 24] suggest adopting Attentions instead of Convolutions because Convolutions can easily learn identical mappings. Nevertheless, both operations are in danger of forming identity mapping by over-concentrating on corresponding input locations for producing the outputs:

$$\text{Conv Kernel} = \begin{bmatrix} 0 & 0 & 0 \\ 0 & 1 & 0 \\ 0 & 0 & 0 \end{bmatrix} , \qquad \text{Attention Map} = \begin{bmatrix} 1 & 0 & 0 & 0 \\ 0 & 1 & 0 & 0 \\ 0 & 0 & 1 & 0 \\ 0 & 0 & 0 & 1 \end{bmatrix} .$$

In Dinomaly, we turn to leverage the "unfocusing ability" of Linear Attention. In order to probe how Attentions propagate information, we train two variants of Dinomaly using vanilla Softmax Attention or Linear Attention as the spatial mixer in the decoder and visualize their attention maps. As shown in Figure 3, Softmax Attention tends to focus on the exact region of the query, while Linear Attention spreads its attention across the whole image. This implies that Linear Attention, forced by its incompetence to focus, utilizes more long-range information to restore features at each position, reducing the chance of passing identical information of unseen patterns to the next layer

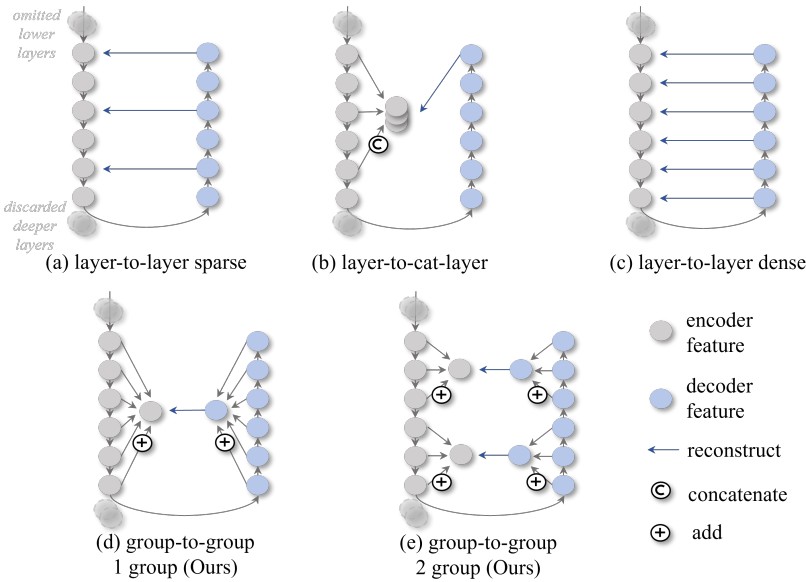

(a) layer-to-layer sparse

(b) layer-to-cat-layer

(c) layer-to-layer dense

(d) group-to-group
1 group (Ours)

(e) group-to-group
2 group (Ours)

encoder
feature

decoder
feature

reconstruct

concatenate

add

Figure 4: Schemes of reconstruction constraint. (a) Layer-to-layer (sparse). (b) Layer-to-cat-layer. (c) Layer-to-layer (dense). (d) Loose group-to-group, 1-group (Ours). (e) Loose group-to-group, 2-group (Ours).

during reconstruction. Of course, employing Linear Attention also benefits from less computation, free of performance drop.

## 2.4 Loose Reconstruction

*"The tighter you squeeze, the less you have."*

**Loose Constraint.** Pioneers of feature-reconstruction/distillation UAD methods [5; 2] are inspired by knowledge distillation [45]. Most reconstruction-based methods distill specific encoder layers (e.g. 3 last layers of 3 ResNet stages) by the corresponding decoder layers [2; 5; 17] (Figure 4(a)) or the last decoder layer [3; 4] (Figure 4(b)). Intuitively, with more encoder-decoder feature pairs (Figure 4(c)), UAD model can utilize more information in different layers to discriminate anomalies. However, according to the intuition of knowledge distillation, the student (decoder) can better mimic the behavior of the teacher (encoder) given more layer-to-layer supervision, which is harmful for UAD models that detect anomalies by encoder-decoder discrepancy. This phenomenon is also embodied as identity mapping. Thanks to the top-to-bottom consistency of columnar Transformer layers, we propose to loosen the layer-to-layer constraint by adding up all feature maps of interested layers as a whole group, as shown in Figure 4(d). This scheme can be seen as loosening the layer-to-layer correspondence, so that the decoder is allowed to act much more differently from the encoder when the input pattern is unseen. Because features of shallow layers contain low-level visual characters that are helpful for precise localization, we can further group the features into the low-semantic-level group and high-semantic-level group, as shown in Figure 4(e).

**Loose Loss.** Following the above analysis, we also loosen the point-by-point reconstruction loss function by discarding some points in the feature map. Here, we simply borrow the hard-mining global cosine loss [18] that detaches the gradients of well-restored feature points with low cosine distance during training. Let $f_E$ and $f_D$ denotes (grouped) feature maps of encoder and decoder:

$$\mathcal{L}_{global-hm} = \mathcal{D}_{cos}(\mathcal{F}(f_E), \mathcal{F}(f_D)) = 1 - \frac{\mathcal{F}(f_E)^T \cdot \mathcal{F}(f_D)}{\|\mathcal{F}(f_E)\| \ \|\mathcal{F}(f_D)\|}, \tag{4}$$

$$f_D(h, w) = \begin{cases} sg(f_D(h, w))_{0.1}, & \text{if } \mathcal{D}_{cos}(f_D(h, w), f_E(h, w)) < 90\% \left[\mathcal{D}_{cos}(f_D, f_E)\right]_{batch} \\ f_D(h, w), & \text{else} \end{cases}, \tag{5}$$

where $\mathcal{F}(\cdot)$ denotes flatten operation, $f_D(h,w)$ represents the feature point at $(h,w)$, $sg(\cdot)_{0.1}$ denotes shrink the gradient to one-tenth of the original [2], $\mathcal{D}_{cos}(f_D(h,w), f_E(h,w)) <$ $90\%[\mathcal{D}_{cos}(f_D, f_E)]_{batch}$ selects 90% feature points with smaller cosine distance within a batch. Total loss is the mean $\mathcal{L}_{global-hm}$ of all encoder-decoder feature pairs.

## 3 Experiments

### 3.1 Experimental Settings

**Datasets.** **MVTec-AD** [20] contains 15 objects (5 texture classes and 10 object classes) with a total of 3,629 normal images as the training set and 1,725 images as the test set (467 normal, 1258 anomalous). **VisA** [21] contains 12 objects. Training and test sets are split following the official splitting, resulting in 8,659 normal images in the training set and 2,162 images in the test set (962 normal, 1,200 anomalous). **Real-IAD** [22] is a large UAD dataset recently released, containing 30 distinct objects. We follow the official splitting that includes all views, resulting in 36,465 normal images in the training set and 114,585 images in the test set (63,256 normal, 51,329 anomalous).

**Metrics.** Following prior works [19; 17], we adopt 7 evaluation metrics. Image-level anomaly detection performance is measured by the Area Under the Receiver Operator Curve (AUROC), Average Precision (AP), and $F_1$ score under optimal threshold ($F_1$-max). Pixel-level anomaly localization is measured by AUROC, AP, $F_1$-max and the Area Under the Per-Region-Overlap (AUPRO). The results of a dataset is the average of all classes.

**Implementation Details.** ViT-Base/14 (patchsize=14) pre-trained by DINOv2-R [39] is used as the encoder by default. The drop rate of Noisy Bottleneck is 0.2 by default and increases to 0.4 on the diverse Real-IAD. Loose constraint with 2 groups is employed, and the anomaly map is given by the mean per-point cosine distance of the 2 groups. The input image is first resized to $448^2$ and then center-cropped to $392^2$, so the feature map ($28^2$) is large enough for anomaly localization. StableAdamW optimizer [46] with AMSGrad [47] (more stable than AdamW [48] in training) is utilized with $lr$=2e-3, $\beta$=(0.9,0.999) and $wd$=1e-4. The network is trained for 10,000 iterations (steps) on MVTec-AD and VisA, and 50,000 iterations on Real-IAD. More details are available in Appendix A.2.

### 3.2 Comparison to Multi-Class UAD SoTAs

We compare the proposed Dinomaly with the most advanced methods. Among them, RD4AD [2] based on feature reconstruction, SimpleNet [13] based on feature-level pseudo-anomaly, and DeST-Seg [12] based on feature reconstruction & pseudo anomaly are designed for conventional class-separated UAD settings. UniAD based on feature reconstruction, ReContrast [18] based on contrastive reconstruction, ViTAD [17] based on feature reconstruction & Transformer, DiAD [49] based on Diffusion reconstruction, and MambaAD [19] based on feature reconstruction & Mamba are designed for MUAD settings. Notably, ViTAD and MambaAD are contemporary *arxiv* preprints released within months. The intuitive comparison is already presented in Figure 1.

Experimental results are presented in Table 1, where Dinomaly surpasses compared methods by a large margin on all datasets and all metrics. On the most widely used MVTec-AD, Dinomaly produces image-level performance of **99.6/99.8/99.0** and pixel-level performance of **98.4/69.3/69.2/94.8**, outperforming previous SoTAs by *1.0/0.2/1.2* and *0.7/9.1/7.7/1.6*. This result declares that the image-level performance on the MVTec-AD dataset is nearly saturated under the MUAD setting. On the popular VisA, Dinomaly achieves image-level performance of **98.7/98.9/96.2** and pixel-level performance of **98.7/53.2/55.7/94.5**, outperforming previous SoTAs by *3.2/2.5/4.2* and *0.2/5.3/5.1/2.6*. On the Real-IAD that contains 30 classes, each with 5 camera views, we produce image-level and pixel-level performance of **89.3/86.8/80.2** and **98.8/42.8/47.1/93.9**, outperforming previous SoTAs by *3.0/2.2/3.2* and *0.3/4.9/5.4/3.4*, indicating our scalability to extremely complex scenarios. Per-class performances and qualitative visualization are presented in Appendix A.5 and A.6. In addition, adopting a larger backbone further improves the above performances, as presented in Table A2.

---

[2]Complete stop-gradient causes optimization instability occasionally.

Table 1: Performance under **multi-class** UAD setting (%). †: method designed for MUAD.

| Dateset | Method | Image-level | | | Pixel-level | | | |
|---|---|---|---|---|---|---|---|---|
| | | AUROC | AP | $F_1$-max | AUROC | AP | $F_1$-max | AUPRO |
| MVTec-AD [20] | RD4AD [2] | 94.6 | 96.5 | 95.2 | 96.1 | 48.6 | 53.8 | 91.1 |
| | SimpleNet [13] | 95.3 | 98.4 | 95.8 | 96.9 | 45.9 | 49.7 | 86.5 |
| | DeSTSeg [12] | 89.2 | 95.5 | 91.6 | 93.1 | 54.3 | 50.9 | 64.8 |
| | UniAD [3]† | 96.5 | 98.8 | 96.2 | 96.8 | 43.4 | 49.5 | 90.7 |
| | ReContrast [18]† | 98.3 | 99.4 | 97.6 | 97.1 | 60.2 | 61.5 | 93.2 |
| | DiAD [49]† | 97.2 | 99.0 | 96.5 | 96.8 | 52.6 | 55.5 | 90.7 |
| | ViTAD [17]† | 98.3 | 99.4 | 97.3 | 97.7 | 55.3 | 58.7 | 91.4 |
| | MambaAD [19]† | 98.6 | 99.6 | 97.8 | 97.7 | 56.3 | 59.2 | 93.1 |
| | **Dinomaly** (Ours) | **99.6** | **99.8** | **99.0** | **98.4** | **69.3** | **69.2** | **94.8** |
| VisA [21] | RD4AD [2] | 92.4 | 92.4 | 89.6 | 98.1 | 38.0 | 42.6 | 91.8 |
| | SimpleNet [13] | 87.2 | 87.0 | 81.8 | 96.8 | 34.7 | 37.8 | 81.4 |
| | DeSTSeg [12] | 88.9 | 89.0 | 85.2 | 96.1 | 39.6 | 43.4 | 67.4 |
| | UniAD [3]† | 88.8 | 90.8 | 85.8 | 98.3 | 33.7 | 39.0 | 85.5 |
| | ReContrast [18]† | 95.5 | 96.4 | 92.0 | 98.5 | 47.9 | 50.6 | 91.9 |
| | DiAD [49]† | 86.8 | 88.3 | 85.1 | 96.0 | 26.1 | 33.0 | 75.2 |
| | ViTAD [17]† | 90.5 | 91.7 | 86.3 | 98.2 | 36.6 | 41.1 | 85.1 |
| | MambaAD [19]† | 94.3 | 94.5 | 89.4 | 98.5 | 39.4 | 44.0 | 91.0 |
| | **Dinomaly** (Ours) | **98.7** | **98.9** | **96.2** | **98.7** | **53.2** | **55.7** | **94.5** |
| Real-IAD [22] | RD4AD [2] | 82.4 | 79.0 | 73.9 | 97.3 | 25.0 | 32.7 | 89.6 |
| | SimpleNet [13] | 57.2 | 53.4 | 61.5 | 75.7 | 2.8 | 6.5 | 39.0 |
| | DeSTSeg [12] | 82.3 | 79.2 | 73.2 | 94.6 | 37.9 | 41.7 | 40.6 |
| | UniAD [3]† | 83.0 | 80.9 | 74.3 | 97.3 | 21.1 | 29.2 | 86.7 |
| | ReContrast [18]† | 86.4 | 84.2 | 77.4 | 97.8 | 31.6 | 38.2 | 91.8 |
| | DiAD [49]† | 75.6 | 66.4 | 69.9 | 88.0 | 2.9 | 7.1 | 58.1 |
| | ViTAD [17]† | 82.3 | 79.4 | 73.4 | 96.9 | 26.7 | 34.9 | 84.9 |
| | MambaAD [19]† | 86.3 | 84.6 | 77.0 | 98.5 | 33.0 | 38.7 | 90.5 |
| | **Dinomaly** (Ours) | **89.3** | **86.8** | **80.2** | **98.8** | **42.8** | **47.1** | **93.9** |

Table 2: Performance under conventional **class-separated** UAD setting (%). n/a: not available.

| Method | MVTec-AD [20] | | | VisA [21] | | | Real-IAD [22] | | |
|---|---|---|---|---|---|---|---|---|---|
| | I-AUROC | P-AP | P-AUPRO | I-AUROC | P-AP | P-AUPRO | I-AUROC | P-AP | P-AUPRO |
| *Dinomaly (MUAD)* | *99.6* | *69.3* | *94.8* | *98.7* | *53.2* | *94.5* | *89.3* | *42.8* | *93.9* |
| **Dinomaly** | **99.7** | 68.9 | **95.0** | **98.9** | 50.7 | **95.1** | **92.0** | **45.2** | **95.1** |
| RD4AD [2] | 98.5 | 58.0 | 93.9 | 96.0 | 27.7 | 70.9 | 87.1 | n/a | 93.8 |
| PatchCore [8] | 99.1 | 56.1 | 93.5 | 95.1 | 40.1 | 91.2 | 89.4 | n/a | 91.5 |
| SimpleNet [13] | 99.6 | 54.8 | 90.0 | 96.8 | 36.3 | 88.7 | 88.5 | n/a | 84.6 |
| EfficientAD [15] | 99.1 | 63.8 | 93.5 | 98.1 | 40.8 | 94.0 | n/a | n/a | n/a |

### 3.3 Comparison to Class-Separated UAD SoTAs

We also compare our Dinomaly with class-separated SoTAs, as shown in Table 2. On MVTec-AD and VisA, our Dinomaly under MUAD setting is comparable to conventional SoTAs that build individual models for each class [2; 13; 8; 15]. In addition, Dinomaly is subjected to nearly no performance drop compared to its class-separated counterpart on these datasets. On the complicated Real-IAD that involves more images, classes, and views, class-separated Dinomaly sets new SoTA records. Multi-class Dinomaly presents moderate performance drop but is still comparable to class-separated SoTAs.

### 3.4 Ablation Study

**Overall Ablation.** We conduct experiments to verify the effectiveness of the proposed elements, i.e., Noisy Bottleneck (NB), Linear Attention (LA), Loose Constraint (LC), and Loose Loss (LL). The already-powerful baseline is Dinomaly with noiseless MLP bottleneck, Softmax Attention, dense layer-to-layer supervision, and global cosine loss [18]. Results on MVTec-AD and VisA are shown in Table 3 and Table A1, respectively. NB and LL can directly contribute to the model performance. LA and LC boost the performance with the presence of NB. The use of LC is not solely beneficial because

Table 3: Ablations of Dinomaly elements on MVTec-AD (%). NB: Noisy Bottleneck. LA: Linear Attention. LC: Loose Constraint (2 groups). LL: Loose Loss. Results on VisA see Table A1.

| NB | LA | LC | LL | Image-level | | | Pixel-level | | | |
|---|---|---|---|---|---|---|---|---|---|---|
| | | | | AUROC | AP | $F_1$-max | AUROC | AP | $F_1$-max | AUPRO |
| | | | | 98.41 | 99.09 | 97.41 | 97.18 | 62.96 | 63.82 | 92.95 |
| ✓ | | | | 99.06 | 99.54 | 98.31 | 97.62 | 66.22 | 66.70 | 93.71 |
| | ✓ | | | 98.54 | 99.21 | 97.62 | 97.20 | 62.94 | 63.73 | 93.09 |
| | | ✓ | | 98.35 | 99.04 | 97.43 | 97.10 | 61.05 | 62.73 | 92.60 |
| | | | ✓ | 99.03 | 99.45 | 98.19 | 97.62 | 64.10 | 64.96 | 93.34 |
| ✓ | ✓ | | | 99.27 | 99.62 | 98.63 | 97.85 | 67.36 | 67.33 | 94.16 |
| ✓ | | ✓ | | 99.50 | 99.72 | 98.87 | 98.14 | 68.16 | 68.24 | 94.23 |
| ✓ | | ✓ | ✓ | 99.52 | 99.73 | 98.92 | 98.20 | 68.25 | 68.34 | 94.17 |
| ✓ | ✓ | ✓ | | 99.57 | **99.78** | 99.00 | 98.20 | 67.93 | 68.21 | 94.50 |
| ✓ | ✓ | ✓ | ✓ | **99.60** | **99.78** | **99.04** | **98.35** | **69.29** | **69.17** | **94.79** |

Table 4: Ablations of Dropout rates in Noisy Bottleneck, conducted on MVTec-AD (%). †: default.

| Dropout rate | Image-level | | | Pixel-level | | | |
|---|---|---|---|---|---|---|---|
| | AUROC | AP | $F_1$-max | AUROC | AP | $F_1$-max | AUPRO |
| 0 (noiseless) | 98.19 | 99.55 | 98.51 | 97.55 | 63.11 | 64.39 | 93.33 |
| 0.1 | 99.54 | 99.75 | 98.90 | **98.35** | **69.46** | **69.19** | 94.53 |
| 0.2 † | 99.60 | 99.78 | 99.04 | **98.35** | 69.29 | 69.17 | **94.79** |
| 0.3 | **99.65** | **99.83** | 99.16 | 98.34 | 68.46 | 68.81 | 94.63 |
| 0.4 | 99.64 | 99.80 | **99.23** | 98.22 | 67.95 | 68.33 | 94.57 |
| 0.5 | 99.56 | 99.81 | 99.14 | 98.15 | 67.43 | 67.82 | 94.64 |

Table 5: Ablations of reconstruction constraint, conducted on MVTec-AD (%). †: default.

| Constraints | Image-level | | | Pixel-level | | | |
|---|---|---|---|---|---|---|---|
| | AUROC | AP | $F_1$-max | AUROC | AP | $F_1$-max | AUPRO |
| layer-to-layer (dense, every 1) | 99.39 | 99.68 | 98.73 | 98.12 | 68.55 | 68.63 | 94.28 |
| layer-to-layer (sparse, every 2) | 99.52 | 99.73 | 98.95 | 98.16 | 68.89 | 68.57 | 94.40 |
| layer-to-layer (sparse, every 4) | 99.54 | 99.77 | 99.05 | 98.04 | 66.69 | 67.17 | 94.07 |
| layer-to-cat-layer (every 2) | 99.48 | 99.71 | 99.26 | 97.83 | 62.29 | 62.91 | 93.16 |
| group-to-group (1 group) | **99.64** | **99.80** | **99.36** | 98.18 | 64.79 | 65.40 | 93.96 |
| group-to-group (2 groups)† | 99.60 | 99.78 | 99.04 | **98.35** | **69.29** | **69.17** | **94.79** |

LC makes the reconstruction too easy without injected noise. Combining some of the proposed elements boosts the performance of the baseline, while employing them all produces the best results. **Noisy Rates**. We conduct ablations on the discarding rate of the Dropouts in MLP bottleneck, as shown in Table 4. Experimental results demonstrate that Dinomaly is robust to different levels of dropout rate. **Reconstruction Constraint.** We quantitatively examine different reconstruction schemes presented in Figure 4. As shown in Table 5, group-to-group LC outperforms layer-to-layer supervision. On image-level metrics, 1-group LC with all layers added performs similarly to its 2-group counterpart that separates low-level and high-level layers; however, 1-group LC mixes low-level and high-level features which is harmful for anomaly localization. More ablations on scalability, input size, pre-trained foundations, etc., are presented in Appendix A.3.

## 4 Conclusion

Dinomaly, a minimalistic UAD framework, is proposed to address the under-performed MUAD models in this paper. We present four key elements in Dinomaly, i.e., Foundation Transformer, Noisy MLP Bottleneck, Linear Attention, and Loose Reconstruction, that can boost the performance under the challenging MUAD setting without fancy modules and tricks. Extensive experiments on MVTec AD, VisA, and Real-IAD demonstrate our superiority over previous model-unified multi-class models and even recent class-separated models, indicating the feasibility of implementing a unified model in complicated scenarios free of severe performance degradation.

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

# A Appendix / supplemental material

## A.1 Related Work

**Epistemic methods** are based on the assumption that the networks respond differently during inference between seen input and unseen input. Within this paradigm, *pixel reconstruction* methods assume that the networks trained on normal images can reconstruct anomaly-free regions well, but poorly for anomalous regions. Auto-encoder (AE) [50; 51], variational auto-encoder (VAE) [52; 53], or generative adversarial network (GAN) [54; 55] are used to restore normal pixels. However, *pixel reconstruction* models may also succeed in restoring unseen anomalous regions if they resemble normal regions in pixel values or the anomalies are barely noticeable [2]. Therefore, *feature reconstruction* is proposed to construct features of pre-trained encoders instead of raw pixels [2; 3; 4]. To prevent the whole network from converging to a trivial solution, the parameters of the encoders are frozen during training. In *feature distillation* [5; 6], the student network is trained from scratch to mimic the output features of the pre-trained teacher network with the same input of normal images, also based on the similar hypothesis that the student trained on normal samples only succeed in mimicking features of normal regions.

**Pseudo-anomaly** methods generate handcrafted defects on normal images to imitate anomalies, converting UAD to supervised classification [11] or segmentation tasks [10]. Specifically, CutPaste [11] simulates anomalous regions by randomly pasting cropped patches of normal images. DRAEM [10] constructs abnormal regions using Perlin noise as the mask and another image as the additive anomaly. DeTSeg [12] employs a similar anomaly generation strategy and combines it with feature reconstruction. SimpleNet [13] introduces anomaly by injecting Gaussian noise in the pre-trained feature space. These methods deeply rely on how well the pseudo anomalies match the real anomalies, which makes it hard to generalize to different datasets.

**Feature statistics** methods [7; 8; 56; 9] memorize all normal features (or their modeled distribution) extracted by networks pre-trained on large-scale datasets and match them with test samples during inference. Since these methods require memorizing, processing, and matching nearly all features from training samples, they are computationally expensive in both training and inference, especially when the training set is large.

**Multi-Class UAD**. UniAD [3] first introduced multi-class anomaly detection, aiming to detect anomalies for different classes using a unified model. In this setting, conventional UAD methods often face the challenge of "identical shortcuts", where both anomaly-free and anomaly samples can be effectively recovered during inference [3]. It is caused by the diversity of multi-class normal patterns that drive the network to generalize on unseen patterns. This contradicts the fundamental assumption of epistemic methods. Many current researches focus on addressing this challenge [3; 24; 18; 57; 25]. UniAD [3] employs a neighbor-masked attention module and a feature-jitter strategy to mitigate these shortcuts. HVQ-Trans [24] proposes a vector quantization (VQ) Transformer model that induces large feature discrepancies for anomalies. LafitE [25] utilizes a latent diffusion model and introduces a feature editing strategy to alleviate this issue. DiAD [26] also employs diffusion models to address multi-class UAD settings. OmniAL [23] focuses on anomaly localization in the unified setting, preventing identical reconstruction by using synthesized pseudo anomalies. ViTAD [58] abstracts a unified feature-reconstruction UAD framework and employ Transformer building blocks. MambaAD [19] explores the recently proposed State Space Model (SSM), Mamba, in the context of multi-class UAD.

**Scope of Application.** In this work, we focus on **sensory AD** that detects regional or structural anomalies (common in practical applications such as industrial inspection, medical disease screening, etc.), which is distinguished from **semantic AD**. In sensory AD, normal and anomalous samples are the same objects except for anomaly, e.g. good cable vs. spoiled cable. In semantic AD, the class of normal samples and anomalous samples are semantically different, e.g. animals vs. vehicles. Semantic AD methods usually utilize and compare the global representation of images, which generally do not suffer from the issues of multi-class setting discussed in this paper..

## A.2 Full Implementation Details

ViT-Base/14 (patch size=14) pre-trained by DINOv2 with registers (DINOv2-R) [39] is utilized as the encoder. The discard rate of Dropout in Noisy Bottleneck is 0.2 by default, which is increased to

0.4 for the diverse Real-IAD. Loose constraint with 2 groups and $\mathcal{L}_{global-hm}$ loss are used by default. The input image is first resized to $448^2$ and then center-cropped to $392^2$, so that the feature map ($28^2$) is large enough for localization. StableAdamW optimizer [46] with AMSGrad [47] is utilized with $lr$ (learning rate)=2e-3, $\beta$=(0.9,0.999), $wd$ (weight decay)=1e-4 and $eps$=1e-10. The network is trained for 10,000 iterations for MVTec-AD and VisA and 50,000 iterations for Real-IAD under MUAD setting. The network is trained for 5,000 iterations on each class under the class-separated UAD setting. The $lr$ warms up from 0 to 2e-3 in the first 100 iterations and cosine anneals to 2e-4 throughout the training. The discarding rate in Equation 5 linearly rises from 0% to 90% in the first 1,000 iterations as warm-up (500 iters for class-separated setting). The anomaly map is obtained by upsampling the point-wise cosine distance between encoder and decoder feature maps (averaging if more than one pair or group). The mean of the top 1% pixels in an anomaly map is used as the image anomaly score. All experiments are conducted with random seed=1 with cuda deterministic for invariable weight initialization and batch order. Codes are implemented with Python 3.8 and PyTorch 1.12.0 cuda 11.3, and run on NVIDIA GeForce RTX3090 GPUs (24GB).

### A.3 Additional Ablation Studies and Experiments

**Ablations on VisA.** Similar to Table 3 that conduct ablation experiments on MVTec-AD, we additionally run them on VisA for further validations. As shown in Table A1, proposed components of Dinomaly contribute to the AD performances on VisA as on MVTec-AD.

Table A1: Ablations of Dinomaly elements on VisA (%). NB: Noisy Bottleneck. LA: Linear Attention. LC: Loosen Constraint (2 groups). LL: Loosen Loss.

| NB | LA | LC | LL | Image-level | | | Pixel-level | | | |
|---|---|---|---|---|---|---|---|---|---|---|
| | | | | AUROC | AP | $F_1$-max | AUROC | AP | $F_1$-max | AUPRO |
| | | | | 95.81 | 96.35 | 92.06 | 97.97 | 47.88 | 52.55 | 93.43 |
| ✓ | | | | 97.38 | 97.74 | 94.07 | 97.84 | 50.42 | 54.57 | 93.71 |
| | ✓ | | | 95.74 | 96.23 | 91.87 | 98.01 | 47.89 | 52.58 | 93.34 |
| | | ✓ | | 96.39 | 97.01 | 92.54 | 97.37 | 46.80 | 51.66 | 92.75 |
| | | | ✓ | 96.93 | 97.26 | 93.32 | 98.37 | 49.52 | 53.59 | 94.11 |
| ✓ | ✓ | | | 97.52 | 97.75 | 94.33 | 98.06 | 51.49 | 55.09 | 93.75 |
| ✓ | | ✓ | | 98.06 | 98.37 | 95.18 | 98.21 | 51.43 | 54.89 | 93.94 |
| ✓ | | ✓ | ✓ | 98.57 | 98.77 | 95.75 | 98.57 | 52.29 | 55.38 | 94.28 |
| ✓ | ✓ | ✓ | | 98.22 | 98.43 | 95.27 | 98.51 | 53.11 | 55.48 | 94.24 |
| ✓ | ✓ | ✓ | ✓ | **98.73** | **98.87** | **96.18** | **98.74** | **53.23** | **55.69** | **94.50** |

**Scalability.** Previous works [3; 2; 17] reported that AD methods do not follow the model "scaling law", i.e., larger models do not necessarily produce better performance. For example, RD4AD [2] found WideResNet50 better than WideResNet101 as the encoder backbone. ViTAD [17] found ViT-Small better than ViT-Base. We conduct experiments to probe the influence of the scale of backbone Transformers in Dinomaly. ViT-Small, ViT-Base (default), and ViT-Large pre-trained by DINOv2-R are used as the encoder, respectively. ViT-Small has 12 layers, so we take the [3,4,5,...10]$th$ layer as the interested 8 middle layers, which is the same as default ViT-Base. ViT-Large has 24 layers, so we take the [5,7,9,...19]$th$ layer as the interested 8 middle layers. The layer hyperparameters of the decoder, such as embedding dimension and numbers of attention heads, follow the hyperparameters of the corresponding encoder. Other training strategies are identical to default. As shown in Table A2, the MUAD performance of Dinomaly follows the "scaling law". Dinomaly equipped with ViT-Small already produces state-of-the-art results. ViT-Large further boosts Dinomaly to an unprecedented higher record.

Table A2: Comparison of different ViT architectures, conducted on MVTec-AD (%). Latency per image is measured on NVIDIA RTX3090 with batch size=16.

| Arch. | Parameters | MACs | Latency | Image-level | | | Pixel-level | | | |
|---|---|---|---|---|---|---|---|---|---|---|
| | | | | AUROC | AP | $F_1$-max | AUROC | AP | $F_1$-max | AUPRO |
| ViT-Small | 37.4M | 26.3G | 6.8ms | 99.26 | 99.67 | 98.72 | 98.07 | 68.29 | 67.78 | 94.36 |
| ViT-Base | 148.0M | 104.7G | 17.2ms | 99.60 | 99.78 | 99.04 | 98.35 | 69.29 | 69.17 | 94.79 |
| ViT-Large | 275.3M | 413.5G | 41.3ms | **99.77** | **99.92** | **99.45** | **98.54** | **70.53** | **70.04** | **95.09** |

**Input Size.** The patch size of ViTs (usually $14 \times 14$ or $16 \times 16$) is much larger than the stem layer's down-sampling rate of CNNs (usually $4 \times 4$), resulting in smaller feature map size. For dense prediction tasks like semantic segmentation, ViTs usually employ a large input image size [30]. This practice holds in anomaly localization as well. In Table A3, we present the results of Dinomaly with different input resolutions. Following PatchCore [8], by default, we adopt center-crop preprocessing to reduce the influence of background, which can also cause unreachable anomalies at the edge of images. Experimental results demonstrate our robustness to input size. While small image size is enough for image-level anomaly detection, larger inputs are beneficial to anomaly localization. All experiments evaluate localization performance in a unified size of $256 \times 256$ for fairness.

Table A3: Ablations of input size, conducted on MVTec-AD (%). $R448^2$-$C392^2$ represents first resizing images to $448 \times 448$, then center cropping to $392 \times 392$.

| Image Size | MACs | Image-level | | | Pixel-level | | | |
|---|---|---|---|---|---|---|---|---|
| | | AUROC | AP | $F_1$-max | AUROC | AP | $F_1$-max | AUPRO |
| $R512^2$-$C448^2$ | 136.4G | **99.67** | 99.81 | 99.12 | 98.33 | _69.24_ | **69.47** | 94.76 |
| $R448^2$ | 136.4G | 99.59 | 99.77 | 99.19 | **98.57** | 68.09 | 68.58 | **95.60** |
| $R448^2$-$C392^2$ | 104.7G | 99.60 | 99.78 | 99.04 | 98.35 | **69.29** | 69.17 | 94.79 |
| $R392^2$ | 104.7G | 99.48 | 99.74 | 99.04 | 98.47 | 67.02 | 67.86 | _95.34_ |
| $R384^2$-$C336^2$ | 77.1G | 99.61 | 99.78 | 99.22 | 98.27 | 67.22 | 67.77 | 94.24 |
| $R336^2$ | 77.1G | _99.63_ | **99.84** | _99.23_ | _98.48_ | 65.46 | 66.60 | 95.10 |
| $R320^2$-$C280^2$ | 53.7G | 99.62 | _99.81_ | 99.07 | 98.21 | 65.21 | 66.34 | 93.57 |
| $R280^2$ | 53.7G | 99.46 | 99.75 | **99.27** | 98.40 | 63.28 | 64.79 | 94.47 |

**Pre-Trained Foundations.** The representation quality of the frozen backbone Transformer is of great significance to unsupervised anomaly detection. We conduct extensive experiments to probe the impact of different pre-training methods, including supervised learning and self-supervised learning. DeiT [59] is trained on ImageNet[1] in a supervised manner by distilling CNNs. MAE [33], BEiTv2 [35], and D-iGPT [60] are based on masked image modeling (MIM). Given input images with masked patches, MAE [33] is optimized to restore raw pixels; BEiTv2 [35] is trained to predict the token index of VQ-GAN and CLIP; D-iGPT [60] is trained to predict the features of CLIP model. DINO [29] is based on positive-pair contrastive learning (CL), which is also referred to as self-distillation. It trains the network to produce similar feature representations given two views (augmentations) of the same image. iBot [36] and DINOv2 [30] combine MIM and CL strategies, marking the SoTA of self-supervised foundation models. DINOv2-R [39] is a variation of DINOv2 that employs 4 extra register tokens.

Table A4: Comparison between pre-trained ViT foundations, conducted on MVTec-AD (%). All models are ViT-Base. The patch size of DINOv2 and DINOv2-R is $14^2$; others are $16^2$.

| Pre-Train Method | Type | Image Size | Image-level | | | Pixel-level | | | |
|---|---|---|---|---|---|---|---|---|
| | | | AUROC | AP | $F_1$-max | AUROC | AP | $F_1$-max | AUPRO |
| DeiT[59] | Supervised | $R512^2$-$C448^2$ | 98.19 | 99.24 | 97.64 | 97.93 | 68.98 | 67.91 | 91.45 |
| MAE[33] | MIM | $R512^2$-$C448^2$ | 96.27 | 98.33 | 95.44 | 96.96 | 62.89 | 63.32 | 89.85 |
| D-iGPT[60] | MIM | $R512^2$-$C448^2$ | 98.75 | 99.24 | 97.70 | 98.30 | 65.77 | 66.16 | 92.34 |
| DINO[29] | CL | $R512^2$-$C448^2$ | 98.97 | 99.58 | 98.14 | 98.52 | 70.89 | 69.02 | 93.48 |
| iBOT[36] | CL+MIM | $R512^2$-$C448^2$ | 99.22 | 99.67 | 98.57 | 98.60 | 70.78 | 69.92 | 93.33 |
| DINOv2[30] | CL+MIM | $R448^2$-$C396^2$ | 99.55 | 99.81 | 99.13 | 98.26 | 68.35 | 68.79 | 94.83 |
| DINOv2-R[39] | CL+MIM | $R448^2$-$C396^2$ | 99.60 | 99.78 | 99.04 | 98.35 | 69.29 | 69.17 | 94.79 |
| DeiT[59] | Supervised | $R256^2$-$C224^2$ | 97.65 | 99.05 | 97.40 | 97.80 | 62.58 | 63.39 | 89.98 |
| MAE[33] | MIM | $R256^2$-$C224^2$ | 97.25 | 98.84 | 96.94 | 97.78 | 63.00 | 64.01 | 90.95 |
| BEiTv2[35] | MIM | $R256^2$-$C224^2$ | 97.70 | 99.11 | 97.39 | 97.61 | 59.79 | 62.53 | 90.10 |
| D-iGPT[60] | MIM | $R256^2$-$C224^2$ | 99.21 | 99.66 | 98.47 | 98.08 | 60.05 | 63.05 | 91.78 |
| DINO[29] | CL | $R256^2$-$C224^2$ | 99.20 | 99.72 | 98.77 | 98.16 | 64.16 | 65.07 | 92.02 |
| iBOT[36] | CL+MIM | $R256^2$-$C224^2$ | 99.31 | 99.74 | 98.77 | 98.25 | 64.01 | 65.37 | 91.68 |
| DINOv2[30] | CL+MIM | $R256^2$-$C224^2$ | 99.26 | 99.70 | 98.60 | 97.95 | 62.27 | 64.39 | 92.80 |
| DINOv2-R[39] | CL+MIM | $R256^2$-$C224^2$ | 99.34 | 99.73 | 99.03 | 98.09 | 63.04 | 64.48 | 92.59 |

It is noted that most models are pre-trained with the image resolution of $224 \times 224$, except that DINOv2 [30] and DINOv2-R [39] have extra a high-resolution training phase with $518 \times 518$. However, directly using the pre-trained weights on a different resolution for UAD without fine-tuning like other supervised tasks can cause generalization problems. Therefore, by default, we still keep the feature size of all compared models to $28 \times 28$, i.e., the input size is $392 \times 392$ for ViT-Base/14 and $448 \times 448$ for ViT-Base/16. Additionally, we train Dinomaly with the low-resolution input size of $224 \times 224$. The results are presented in Table A4. Generally speaking, CL+MIM combined models outperform MIM and CL models. In addition, MIM-based models do not benefit from higher resolutions but suffer from them, indicating the lack of generalization on a different input size. Methods involving CL can better adapt to a higher resolution as they optimize the global representation of class tokens in pre-training, which is insensitive to input size. As expected, DINOv2 and DINOv2-R pre-trained on larger inputs can better benefit from higher resolution in Dinomaly. Because some methods, i.e., D-iGPT, DINO, and iBOT, produce similar results to DINOv2 in $224 \times 224$, we expect that they also have the potential to be as powerful in Dinomaly if they are pre-trained in high-resolution.

**Attention *vs*. Convolution.** Previous works and this paper have proposed to leverage attentions instead of convolutions in UAD. Here, we conduct experiments substituting the attention in the decoder of Dinomaly by convolutions as the spatial mixers. Following MetaFormer [61], we employ Inverted Bottleneck block that consists of $1 \times 1$ conv, GELU activation, $N \times N$ deep-wise conv, and $1 \times 1$ conv, sequentially. The results are shown in Table A5, where Attentions outperform Convolutions, especially for pixel-level anomaly localization. In addition, utilizing convolutions in the decoder can still yield SoTA results, demonstrating the universality of the proposed Dinomaly.

**Neighbour-Masking.** Prior method [3] proposed to mask the keys and values in an $n \times n$ square centered at each query, in order to alleviate identity mapping in Attention. This mechanism can also be applied to Linear Attention as well. As shown in Table A5, neighbor-masking can further improve Dinomaly with both Softmax Attention and Linear Attention moderately.

Table A5: Comparison between Convolutional block, Softmax Attention, and Linear Attention as the spatial mixer of decoder, conducted on MVTec-AD (%).

| Spatial Mixer | Image-level | | | Pixel-level | | | |
|---|---|---|---|---|---|---|---|
| | AUROC | AP | $F_1$-max | AUROC | AP | $F_1$-max | AUPRO |
| ConvBlock $3 \times 3$ | 99.45 | 99.63 | 98.64 | 98.05 | 65.35 | 68.07 | 94.17 |
| ConvBlock $5 \times 5$ | 99.41 | 99.62 | 98.86 | 97.99 | 66.64 | 67.47 | 94.24 |
| ConvBlock $7 \times 7$ | 99.42 | 99.65 | 98.86 | 98.01 | 67.57 | 67.94 | 94.45 |
| Softmax Attention | 99.52 | 99.73 | 98.92 | 98.20 | 68.25 | 68.34 | 94.17 |
| Softmax Attention w/ Neighbour-Mask $n = 1$ | 99.51 | 99.71 | 98.90 | 98.17 | 67.86 | 67.92 | 94.27 |
| Softmax Attention w/ Neighbour-Mask $n = 3$ | 99.56 | 99.76 | 99.05 | 98.28 | 69.26 | 68.17 | 94.50 |
| Linear Attention | **99.60** | 99.78 | 99.04 | 98.35 | 69.29 | 69.17 | **94.79** |
| Linear Attention w/ Neighbour-Mask $n = 1$ | **99.60** | 99.78 | 99.04 | 98.32 | 68.77 | 68.72 | 94.75 |
| Linear Attention w/ Neighbour-Mask $n = 3$ | **99.60** | 99.80 | 99.14 | 98.38 | 69.65 | 69.38 | 94.70 |

**Feature Noise.** Prior method [3] proposed to perturb the encoder features by Feature Jitter, i.e. adding Gaussian noise with *scale* to control the noise magnitude. We evaluate the feature jitter strategy in the proposed Dinomaly by placing it at the beginning of Noisy Bottleneck. As shown in Table A6, both Dropout and Feature Jitter can be a good noise injector in Noisy Bottleneck. Meanwhile, Dropout is more robust to the noisy scale hyperparameter, and more elegant without introducing new modules.

**Random Seeds.** Due to limited computation resources, experiments in this paper are conducted for one run with random seed=1. Here, we conduct 5 runs with 5 random seeds on MVTec-AD. As shown in Table A7, Dinomaly is robust to randomness.

## A.4 Limitation

Vision Transformers are known for their high computation cost, which can be a barrier to low-computation scenarios that require inference speed. Future research can be conducted on the efficiency of Transformer-based methods, such as distillation, pruning, and hardware-friendly attention mechanism (such as FlashAttention).

Table A6: Dropout *vs.* feature jitter, conducted on MVTec-AD (%).

| Noise type | Image-level | | | Pixel-level | | | |
|---|---|---|---|---|---|---|---|
| | AUROC | AP | $F_1$-max | AUROC | AP | $F_1$-max | AUPRO |
| No Noise | 98.19 | 99.55 | 98.51 | 97.55 | 63.11 | 64.39 | 93.33 |
| Feature Jitter *scale*=1 | 99.23 | 99.54 | 98.48 | 97.58 | 63.22 | 64.31 | 93.55 |
| Feature Jitter *scale*=5 | 99.24 | 99.57 | 98.55 | 97.84 | 65.28 | 65.81 | 93.75 |
| Feature Jitter *scale*=10 | 99.46 | 99.73 | 99.12 | 98.19 | 67.59 | 67.80 | 94.19 |
| Feature Jitter *scale*=20 | 99.59 | 99.79 | 99.04 | 98.23 | 67.93 | 68.21 | 94.40 |
| Dropout p=0.1 | 99.54 | 99.75 | 98.90 | **98.35** | **69.46** | **69.19** | 94.53 |
| Dropout p=0.2 | 99.60 | 99.78 | 99.04 | **98.35** | 69.29 | 69.17 | **94.79** |
| Dropout p=0.3 | **99.65** | **99.83** | 99.16 | 98.34 | 68.46 | 68.81 | 94.63 |
| Dropout p=0.4 | 99.64 | 99.80 | **99.23** | 98.22 | 67.95 | 68.33 | 94.57 |

Table A7: Results of 5 random seeds on MVTec-AD (%).

| Random Seed | Image-level | | | Pixel-level | | | |
|---|---|---|---|---|---|---|---|
| | AUROC | AP | $F_1$-max | AUROC | AP | $F_1$-max | AUPRO |
| seed=1 | 99.60 | 99.78 | 99.04 | 98.35 | 69.29 | 69.17 | 94.79 |
| seed=2 | 99.63 | 99.79 | 99.12 | 98.33 | 68.73 | 68.91 | 94.63 |
| seed=3 | 99.63 | 99.79 | 99.16 | 98.31 | 68.70 | 68.93 | 94.60 |
| seed=4 | 99.56 | 99.74 | 99.02 | 98.33 | 69.04 | 69.09 | 94.70 |
| seed=5 | 99.59 | 99.77 | 99.02 | 98.32 | 68.64 | 68.47 | 94.51 |
| mean±std | 99.60±0.03 | 99.77±0.02 | 99.07±0.06 | 98.33±0.01 | 68.88±0.25 | 68.91±0.24 | 94.65±0.09 |

As discussed in section A.1, Dinomaly is used for sensory AD that aims to detect regional anomalies in normal backgrounds. It is not suitable for semantic AD. Previous works have shown that methods designed for sensory AD usually fail to be competitive under semantic AD tasks [3; 2]. Conversely, methods designed for semantic AD do not perform well on sensory AD tasks [62; 37]. Future work can be conducted to unify these two tasks, but according to the "no free lunch" theorem, we believe that methods designed for specific anomaly assumption are likely to be more convincing.

Other special UAD settings, such as zero-shot UAD (vision-language model based) [63], few-shot UAD [64], UAD under noisy training set [65], are not included in this work.

## A.5 Results Per-Category

For future research, we report the per-class results of MVTec-AD [20], VisA [21], and Real-IAD [22]. The performance of compared methods is drawn from MambaAD [19]. Thanks for their exhaustive reproducing. The results of image-level anomaly detection and pixel-level anomaly localization on MVTec-AD are presented in Table A8 and Table A9, respectively. The results of image-level anomaly detection and pixel-level anomaly localization on VisA are presented in Table A10 and Table A11, respectively. The results of image-level anomaly detection and pixel-level anomaly localization on Real-IAD are presented in Table A12 and Table A13, respectively.

## A.6 Qualitative Visualization

We visualize the output anomaly maps of Dinomaly on MVTec-AD, VisA, and Real-IAD, as shown in Figure A1, Figure A2, and Figure A3. It is noted that all visualized samples are randomly chosen without artificial selection.

Table A8: Per-class performance on **MVTec-AD** dataset for multi-class anomaly detection with AUROC/AP/$F_1$-max metrics.

| | Category ↓ | RD4AD [2] CVPR'22 | UniAD [3] NeurIPS'22 | SimpleNet [13] CVPR'23 | DeSTSeg [12] CVPR'23 | DiAD [49] AAAI'24 | MambaAD [19] Arxiv'24 | Dinomaly Ours |
|---|---|---|---|---|---|---|---|---|
| Objects | Bottle | 99.6/99.9/98.4 | 99.7/**100.**/**100.** | **100./100./100.** | 98.7/99.6/96.8 | 99.7/96.5/91.8 | **100./100./100.** | **100./100./100.** |
| | Cable | 84.1/89.5/82.5 | 95.2/95.9/88.0 | 97.5/98.5/94.7 | 89.5/94.6/85.9 | 94.8/98.8/95.2 | 98.8/99.2/95.7 | **100./100./100.** |
| | Capsule | 94.1/96.9/96.9 | 86.9/97.8/94.4 | 90.7/97.9/93.5 | 82.8/95.9/92.6 | 89.0/97.5/95.5 | 94.4/98.7/94.9 | **97.9/99.5/97.7** |
| | Hazelnut | 60.8/69.8/86.4 | 99.8/**100.**/99.3 | 99.9/99.9/99.3 | 98.8/99.2/98.6 | 99.5/99.7/97.3 | **100./100./100.** | **100./100./100.** |
| | Metal Nut | **100./100.**/99.5 | 99.2/99.9/99.5 | 96.9/99.3/96.1 | 92.9/98.4/92.2 | 99.1/96.0/91.6 | 99.9/**100.**/99.5 | **100./100./100.** |
| | Pill | 97.5/99.6/96.8 | 93.7/98.7/95.7 | 88.2/97.7/92.5 | 77.1/94.4/91.7 | 95.7/98.5/94.5 | 97.0/99.5/96.2 | **99.1/99.9/98.3** |
| | Screw | 97.7/99.3/95.8 | 87.5/96.5/89.0 | 76.7/90.6/87.7 | 69.9/88.4/85.4 | 90.7/**99.7/97.9** | 94.7/97.9/94.0 | **98.4**/99.5/96.1 |
| | Toothbrush | 97.2/99.0/94.7 | 94.2/97.4/95.2 | 89.7/95.7/92.3 | 71.7/89.3/84.5 | 99.7/99.9/99.2 | 98.3/99.3/98.4 | **100./100./100.** |
| | Transistor | 94.2/95.2/90.0 | 99.8/98.0/93.8 | 99.2/98.7/97.6 | 78.2/79.5/68.8 | 99.8/96.9/97.4 | **100./100./100.** | 99.0/98.0/96.4 |
| | Zipper | 99.5/99.9/99.2 | 95.8/99.5/97.1 | 99.0/99.7/98.3 | 88.4/96.3/93.1 | 95.1/99.1/94.4 | 99.3/99.8/97.5 | **100./100./100.** |
| Textures | Carpet | 98.5/99.6/97.2 | **99.8/99.9/99.4** | 95.7/98.7/93.2 | 95.9/98.8/94.9 | 99.4/99.9/98.3 | **99.8/99.9/99.4** | 99.8/**100.**/98.9 |
| | Grid | 98.0/99.4/96.5 | 98.2/99.5/97.3 | 97.6/99.2/96.4 | 97.9/99.2/96.6 | 98.5/99.8/97.7 | **100./100./100.** | 99.9/**100.**/99.1 |
| | Leather | **100./100./100.** | **100./100./100.** | **100./100./100.** | 99.2/99.8/98.9 | 99.8/99.7/97.6 | **100./100./100.** | **100./100./100.** |
| | Tile | 98.3/99.3/96.4 | 99.3/99.8/98.2 | 99.3/99.8/98.8 | 97.0/98.9/95.3 | 96.8/99.9/98.4 | 98.2/99.3/95.4 | **100./100./100.** |
| | Wood | 99.2/99.8/98.3 | 98.6/99.6/96.6 | 98.4/99.5/96.7 | 99.9/**100.**/99.2 | 99.7/**100./100.** | 98.8/99.6/96.6 | 99.8/99.9/99.2 |
| | Mean | 94.6/96.5/95.2 | 96.5/98.8/96.2 | 95.3/98.4/95.8 | 89.2/95.5/91.6 | 97.2/99.0/96.5 | 98.6/99.6/97.8 | **99.6/99.8/99.0** |

Table A9: Per-class performance on **MVTec-AD** dataset for multi-class anomaly localization with AUROC/AP/$F_1$-max/AUPRO metrics.

| | Category ↓ | RD4AD [2] CVPR'22 | UniAD [3] NeurIPS'22 | SimpleNet [13] CVPR'23 | DeSTSeg [12] CVPR'23 | DiAD [49] AAAI'24 | MambaAD [19] Arxiv'24 | Dinomaly Ours |
|---|---|---|---|---|---|---|---|---|
| Objects | Bottle | 97.8/68.2/67.6/94.0 | 98.1/66.0/69.2/93.1 | 97.2/53.8/62.4/89.0 | 93.3/61.7/56.0/67.5 | 98.4/52.2/54.8/86.6 | 98.8/79.7/76.7/95.2 | **99.2/88.6/84.2/96.6** |
| | Cable | 85.1/26.3/33.6/75.1 | 97.3/39.9/45.2/86.1 | 96.7/42.4/51.2/85.4 | 89.3/37.5/40.5/49.4 | 96.8/50.1/57.8/80.5 | 95.8/42.2/48.1/90.3 | **98.6/72.0/74.3/94.2** |
| | Capsule | **98.8**/43.4/50.0/94.8 | 98.5/42.7/46.5/92.1 | 98.5/35.4/44.3/84.5 | 95.8/47.9/48.9/62.1 | 97.1/42.0/45.3/87.2 | 98.4/43.9/47.7/92.6 | 98.7/**61.4/60.3/97.2** |
| | Hazelnut | 97.9/36.2/51.6/92.7 | 98.1/55.2/56.8/94.1 | 98.4/44.6/51.4/87.4 | 98.2/65.8/61.6/84.5 | 98.3/79.2/**80.4**/91.5 | 99.0/63.6/64.4/95.7 | **99.4/82.2**/76.4/**97.0** |
| | Metal Nut | 94.8/55.5/66.4/91.9 | 62.7/14.6/29.2/81.8 | **98.0/83.1**/79.4/85.2 | 84.2/42.0/22.8/53.0 | 97.3/30.0/38.3/90.6 | 96.7/74.5/79.1/93.7 | 96.9/78.6/**86.7/94.9** |
| | Pill | 97.5/63.4/65.2/95.8 | 95.0/44.0/53.9/95.3 | 96.5/72.4/67.7/81.9 | 96.2/61.7/41.8/27.9 | 95.7/46.0/51.4/89.0 | 97.4/64.0/66.5/95.7 | **97.8/76.4/71.6/97.3** |
| | Screw | 99.4/40.2/44.6/96.8 | 98.3/28.7/37.6/95.2 | 96.5/15.9/23.2/84.0 | 93.8/19.9/25.3/47.3 | 97.9/60.6/59.6/95.0 | 99.5/49.8/50.9/97.1 | **99.6/60.2/59.6/98.3** |
| | Toothbrush | **99.0**/53.6/58.8/92.0 | 98.4/34.9/45.7/87.9 | 98.4/46.9/52.5/87.4 | 96.2/52.9/58.8/30.9 | **99.0**/78.7/72.8/95.0 | **99.0**/48.5/59.2/91.7 | 98.9/51.5/**62.6/95.3** |
| | Transistor | 85.9/42.3/45.2/74.7 | **97.9/59.5/64.6/93.5** | 93.1/31.4/32.2/66.8 | 73.6/38.4/39.2/43.9 | 95.1/15.6/31.7/90.0 | 96.5/69.4/67.1/87.0 | 93.2/59.9/58.5/77.0 |
| | Zipper | 98.5/53.9/60.3/94.1 | 96.8/40.1/49.9/92.6 | 97.9/53.4/54.6/90.7 | 97.3/64.7/59.2/66.9 | 96.2/60.7/60.0/91.6 | 98.4/60.4/61.7/94.3 | **99.2/79.5/75.4/97.2** |
| Textures | Carpet | 99.0/58.5/60.4/95.1 | 98.5/49.9/51.1/94.4 | 97.4/38.7/43.2/90.6 | 93.6/59.9/58.9/89.3 | 98.6/42.2/46.4/90.6 | 99.2/60.0/63.3/96.7 | **99.3/68.7/71.1/97.6** |
| | Grid | 96.5/23.0/28.4/97.0 | 63.1/10.7/11.9/92.9 | 96.8/20.5/27.6/88.6/ | 97.0/42.1/46.9/86.8 | 96.6/66.0/64.1/94.0 | 99.2/47.4/47.7/97.0 | **99.4/55.3/57.7/97.2** |
| | Leather | 99.3/38.0/45.1/97.4 | 98.8/32.9/34.4/96.8 | 98.7/28.5/32.9/92.7 | **99.5/71.5/66.5**/91.1 | 98.8/56.1/62.3/91.3 | 99.4/50.3/53.3/98.7 | 99.4/52.2/55.0/97.6 |
| | Tile | 95.3/48.5/60.5/85.8 | 91.8/42.1/50.6/78.4 | 94.0/37.0/41.0/70.5 | 93.0/71.0/66.2/87.1 | 92.4/65.7/64.1/90.7 | 93.8/45.1/54.8/80.0 | **98.1/80.1/75.7**/90.5 |
| | Wood | 95.3/47.8/51.0/90.0 | 93.2/37.2/41.5/86.7 | 91.4/34.8/39.7/76.3 | 95.9/**77.3/71.3**/83.4 | 93.3/43.3/43.5/**97.5** | 94.4/46.2/48.2/91.2 | **97.6**/72.8/68.4/94.0 |
| | Mean | 96.1/48.6/53.8/91.1 | 96.8/43.4/49.5/90.7 | 96.9/45.9/49.7/86.5 | 93.1/54.3/50.9/64.8 | 96.8/52.6/55.5/90.7 | 97.7/56.3/59.2/93.1 | **98.4/69.3/69.2/94.8** |

Table A10: Per-class performance on **VisA** dataset for multi-class anomaly detection with AUROC/AP/F1-max metrics.

| Category ↓ | RD4AD [2] CVPR'22 | UniAD [3] NeurIPS'22 | SimpleNet [13] CVPR'23 | DeSTSeg [12] CVPR'23 | DiAD [49] AAAI'24 | MambaAD Arxiv'24 | Dinomaly Ours |
|---|---|---|---|---|---|---|---|
| pcb1 | 96.2/95.5/91.9 | 92.8/92.7/87.8 | 91.6/91.9/86.0 | 87.6/83.1/83.7 | 88.1/88.7/80.7 | 95.4/93.0/91.6 | **99.1/99.1/96.6** |
| pcb2 | 97.8/97.8/94.2 | 87.8/87.7/83.1 | 92.4/93.3/84.5 | 86.5/85.8/82.6 | 91.4/91.4/84.7 | 94.2/93.7/89.3 | **99.3/99.2/97.0** |
| pcb3 | 96.4/96.2/91.0 | 78.6/78.6/76.1 | 89.1/91.1/82.6 | 93.7/95.1/87.0 | 86.2/87.6/77.6 | 93.7/94.1/86.7 | **98.9/98.9/96.1** |
| pcb4 | 99.9/99.9/99.0 | 98.8/98.8/94.3 | 97.0/97.0/93.5 | 97.8/97.8/92.7 | 99.6/99.5/97.0 | **99.9/99.9/98.5** | 99.8/99.8/98.0 |
| macaroni1 | 75.9/ 1.5/76.8 | 79.9/79.8/72.7 | 85.9/82.5/73.1 | 85.7/85.2/78.4 | 76.6/89.0/71.0 | 91.6/89.8/81.6 | **98.0/97.6/94.2** |
| macaroni2 | 88.3/84.5/83.8 | 71.6/71.6/69.9 | 68.3/54.3/59.7 | 68.9/62.1/67.7 | 62.5/57.4/69.6 | 81.6/78.0/73.8 | **95.9/95.7/90.7** |
| capsules | 82.2/90.4/81.3 | 55.6/55.6/76.9 | 74.1/82.8/74.6 | 87.1/93.0/84.2 | 58.2/69.0/78.5 | 91.8/95.0/88.8 | **98.6/99.0/97.1** |
| candle | 92.3/92.9/86.0 | 94.1/94.0/86.1 | 84.1/73.3/76.6 | 94.9/94.8/89.2 | 92.8/92.0/87.6 | 96.8/96.9/90.1 | **98.7/98.8/95.1** |
| cashew | 92.0/95.8/90.7 | 92.8/92.8/91.4 | 88.0/91.3/84.7 | 88.0/92.6/88.1 | 91.5/95.7/89.7 | 94.5/97.3/91.1 | **98.7/99.4/97.0** |
| chewinggum | 94.9/97.5/92.1 | 96.3/96.2/95.2 | 96.4/98.2/93.8 | 95.8/98.3/94.7 | 99.1/99.5/95.9 | 97.7/98.9/94.2 | **99.8/99.9/99.0** |
| fryum | 95.3/97.9/91.5 | 83.0/83.0/85.0 | 88.4/93.0/83.3 | 92.1/96.1/89.5 | 89.8/95.0/87.2 | 95.2/97.7/90.5 | **98.8/99.4/96.5** |
| pipe_fryum | 97.9/98.9/96.5 | 94.7/94.7/93.9 | 90.8/95.5/88.6 | 94.1/97.1/91.9 | 96.2/98.1/93.7 | 98.7/99.3/97.0 | **99.2/99.7/97.0** |
| Mean | 92.4/92.4/89.6 | 85.5/85.5/84.4 | 87.2/87.0/81.8 | 88.9/89.0/85.2 | 86.8/88.3/85.1 | 94.3/94.5/89.4 | **98.7/98.9/96.2** |

Table A11: Per-class performance on **VisA** dataset for multi-class anomaly localization with AUROC/AP/$F_1$-max/AUPRO metrics.

| Category ↓ | RD4AD [2] CVPR'22 | UniAD [3] NeurIPS'22 | SimpleNet [13] CVPR'23 | DeSTSeg [12] CVPR'23 | DiAD [49] AAAI'24 | MambaAD Arxiv'24 | Dinomaly Ours |
|---|---|---|---|---|---|---|---|
| pcb1 | 99.4/66.2/62.4/**95.8** | 93.3/ 3.9/ 8.3/64.1 | 99.2/86.1/78.8/83.6 | 95.8/46.4/49.0/83.2 | 98.7/49.6/52.8/80.2 | **99.8**/77.1/72.4/92.8 | 99.5/**87.9/80.5**/95.1 |
| pcb2 | 98.0/22.3/30.0/90.8 | 93.9/ 4.2/ 9.2/66.9 | 96.6/ 8.9/18.6/85.7 | 97.3/14.6/28.2/79.9 | 95.2/ 7.5/16.7/67.0 | **98.9**/13.3/23.4/89.6 | 98.0/**47.0/49.8/91.3** |
| pcb3 | 97.9/26.2/35.2/93.9 | 97.3/13.8/21.9/70.6 | 97.2/31.0/36.1/85.1 | 97.7/28.1/33.4/62.4 | 96.7/ 8.0/18.8/68.9 | **99.1**/18.3/27.4/89.1 | 98.4/**41.7/45.3/94.6** |
| pcb4 | 97.8/31.4/37.0/88.7 | 94.9/14.7/22.9/72.3 | 93.9/23.9/32.9/61.1 | 95.8/**53.0/53.2**/76.9 | 97.0/17.6/27.2/85.0 | 98.6/47.0/46.9/87.6 | **98.7**/50.5/53.1/**94.4** |
| macaroni1 | 99.4/ 2.9/6.9/95.3 | 97.4/ 3.7/ 9.7/84.0 | 98.9/ 3.5/8.4/92.0 | 99.1/ 5.8/13.4/62.4 | 94.1/10.2/16.7/68.5 | 99.5/17.5/27.6/95.2 | **99.6/33.5/40.6/96.4** |
| macaroni2 | 99.7/13.2/21.8/97.4 | 95.2/ 0.9/ 4.3/76.6 | 93.2/ 0.6/ 3.9/77.8 | 98.5/ 6.3/14.4/70.0 | 93.6/ 0.9/ 2.8/73.1 | 99.5/ 9.2/16.1/96.2 | **99.7/24.7/36.1/98.7** |
| capsules | 99.4/60.4/60.8/93.1 | 88.7/ 3.0/ 7.4/43.7 | 97.1/52.9/53.3/73.7 | 96.9/33.2/ 9.1/76.7 | 97.3/10.0/21.0/77.9 | 99.1/61.3/59.8/91.8 | **99.6/65.0/66.6/97.4** |
| candle | 99.1/25.3/35.8/94.9 | 98.5/17.6/27.9/91.6 | 97.6/ 8.4/16.5/87.6 | 98.7/39.9/45.8/69.0 | 97.3/12.8/22.8/89.4 | 99.0/23.2/32.4/**95.5** | **99.4/43.0/47.9**/95.4 |
| cashew | 91.7/44.2/49.7/86.2 | 98.6/51.7/58.3/87.9 | **98.9/68.9/66.0**/84.1 | 87.9/47.6/52.1/66.3 | 90.9/53.1/60.9/61.8 | 94.3/46.8/51.4/87.8 | 97.1/64.5/62.4/**94.0** |
| chewinggum | 98.7/59.9/61.7/76.9 | 97.9/26.8/29.8/78.3 | 98.8/**86.9/81.0**/68.3 | 94.7/11.9/25.8/59.5 | 98.1/57.5/59.9/79.7 | | **99.1**/65.0/67.7/**88.1** |
| fryum | 97.0/47.6/51.5/93.4 | 95.9/34.0/40.6/76.2 | 93.0/39.1/45.4/85.1 | 88.1/35.2/38.5/47.7 | **97.6/58.6/60.1**/81.3 | 96.9/47.8/51.9/91.6 | 96.6/51.6/53.4/**93.5** |
| pipe_fryum | 99.1/56.8/58.8/**95.4** | 98.9/50.2/57.7/91.5 | 98.5/65.6/63.4/83.0 | 98.9/78.8/72.7/45.9 | **99.4/72.7/69.9**/89.9 | 99.1/53.5/58.5/95.1 | 99.2/64.3/65.1/95.2 |
| Mean | 98.1/38.0/42.6/91.8 | 95.9/21.0/27.0/75.6 | 96.8/34.7/37.8/81.4 | 96.1/39.6/43.4/67.4 | 96.0/26.1/33.0/75.2 | 98.5/39.4/44.0/91.0 | **98.7/53.2/55.7/94.5** |

Table A12: Per-class performance on **Real-IAD** dataset for multi-class anomaly detection with AUROC/AP/$F_1$-max metrics.

| Method → Category ↓ | RD4AD [2] CVPR'22 | UniAD [3] NeurIPS'22 | SimpleNet [13] CVPR'23 | DeSTSeg [12] CVPR'23 | DiAD [49] AAAI'24 | MambaAD Arxiv'24 | Dinomaly Ours |
|---|---|---|---|---|---|---|---|
| audiojack | 76.2/63.2/60.8 | 81.4/76.6/64.9 | 58.4/44.2/50.9 | 81.1/72.6/64.5 | 76.5/54.3/65.7 | 84.2/76.5/67.4 | **86.8**/**82.4**/**72.2** |
| bottle cap | 89.5/86.3/81.0 | 92.5/91.7/81.7 | 54.1/47.6/60.3 | 78.1/74.6/68.1 | 91.6/**94.0**/**87.9** | **92.8**/92.0/82.1 | 89.9/86.7/81.2 |
| button battery | 73.3/78.9/76.1 | 75.9/81.6/76.3 | 52.5/60.5/72.4 | **86.7**/**89.2**/**83.5** | 80.5/71.3/70.6 | 79.8/85.3/77.8 | 86.6/88.9/82.1 |
| end cap | 79.8/84.0/77.8 | 80.9/86.1/78.0 | 51.6/60.8/72.9 | 77.9/81.1/77.1 | 85.1/83.4/**84.8** | 78.0/82.8/77.2 | **87.0**/**87.5**/83.4 |
| eraser | 90.0/88.7/79.7 | **90.3**/**89.2**/**80.2** | 46.4/39.1/55.8 | 84.6/82.9/71.8 | 80.0/80.0/77.3 | 87.5/86.2/76.1 | 90.3/87.6/78.6 |
| fire hood | 78.3/70.1/64.5 | 80.6/74.8/66.4 | 58.1/41.9/54.4 | 81.7/72.4/67.7 | 83.3/**81.7**/**80.5** | 79.3/72.5/64.8 | **83.8**/76.2/69.5 |
| mint | 65.8/63.1/64.8 | 67.0/66.6/64.6 | 52.4/50.3/63.7 | 58.4/55.8/63.7 | **76.7**/**76.7**/**76.0** | 70.1/70.8/65.5 | 73.1/72.0/67.7 |
| mounts | 88.6/79.9/74.8 | 87.6/77.3/77.2 | 58.7/48.1/52.4 | 74.7/56.5/63.1 | 75.3/74.5/**82.5** | 86.8/78.0/73.5 | **90.4**/**84.2**/78.0 |
| pcb | 79.5/85.8/79.7 | 81.0/88.2/79.1 | 54.5/66.0/75.5 | 82.0/88.7/79.6 | 86.0/85.1/85.4 | 89.1/93.7/84.0 | **92.0**/**95.3**/**87.0** |
| phone battery | 87.5/83.3/77.1 | 83.6/80.0/71.6 | 51.6/43.8/58.0 | 83.3/81.8/72.1 | 82.3/77.7/75.9 | 90.2/88.9/80.5 | **92.9**/**91.6**/**82.5** |
| plastic nut | 80.3/68.0/64.4 | 80.0/69.2/63.7 | 59.2/40.3/51.8 | 83.1/75.4/66.5 | 71.9/58.2/65.6 | 87.1/80.7/70.7 | **88.3**/**81.8**/**74.7** |
| plastic plug | 81.9/74.3/68.8 | 81.4/75.9/67.6 | 48.2/38.4/54.6 | 71.7/63.1/60.0 | 88.7/**89.2**/**90.9** | 85.7/82.2/72.6 | **90.5**/86.4/78.6 |
| porcelain doll | 86.3/76.3/71.5 | 85.1/75.2/69.3 | 66.3/54.5/52.1 | 78.7/66.2/64.3 | 72.6/66.8/65.2 | **88.0**/**82.2**/**74.1** | 85.1/73.3/69.6 |
| regulator | 66.9/48.8/47.7 | 56.9/41.5/44.5 | 50.5/29.0/43.9 | 79.2/63.5/56.9 | 72.1/71.4/**78.2** | 69.7/58.7/50.4 | **85.2**/**78.9**/69.8 |
| rolled strip base | 97.5/98.7/94.7 | 98.7/99.3/96.5 | 59.0/75.7/79.8 | 96.5/98.2/93.0 | 68.4/55.9/56.8 | 98.0/99.0/95.0 | **99.2**/**99.6**/**97.1** |
| sim card set | 91.6/91.8/84.8 | 89.7/90.3/83.2 | 63.1/69.7/70.8 | 95.5/96.2/**89.2** | 72.6/53.7/61.5 | 94.4/95.1/87.2 | **95.8**/**96.3**/88.8 |
| switch | 84.3/87.2/77.9 | 85.5/88.6/78.4 | 62.2/66.8/68.6 | 90.1/92.8/83.1 | 73.4/49.4/61.2 | 91.7/94.0/85.4 | **97.8**/**98.1**/**93.3** |
| tape | 96.0/95.1/87.6 | **97.2**/**96.2**/**89.4** | 49.9/41.1/54.5 | 94.5/93.4/85.9 | 73.9/57.8/66.1 | 96.8/95.9/89.3 | 96.9/95.0/88.8 |
| terminalblock | 89.4/89.7/83.1 | 87.5/89.1/81.0 | 59.8/64.7/68.8 | 83.1/86.2/76.6 | 62.1/36.4/47.8 | 96.1/96.8/90.0 | **96.7**/**97.4**/**91.1** |
| toothbrush | 82.0/83.8/77.2 | 78.4/80.1/75.6 | 65.9/70.0/70.1 | 83.7/85.3/79.0 | **91.2**/**93.7**/**90.9** | 85.1/86.2/80.3 | 90.4/91.9/83.4 |
| toy | 69.4/74.2/75.9 | 68.4/75.1/74.8 | 57.8/64.4/73.4 | 70.3/74.8/75.4 | 66.2/57.3/59.8 | 83.0/87.5/79.6 | **85.6**/**89.1**/**81.9** |
| toy brick | 63.6/56.1/59.0 | **77.0**/**71.1**/**66.2** | 58.3/44.9/58.2 | 73.2/68.7/63.3 | 68.4/45.3/55.9 | 70.5/63.7/61.6 | 72.3/65.1/63.4 |
| transistor1 | 91.0/94.0/85.1 | 93.7/95.9/88.9 | 62.2/69.2/72.1 | 90.2/92.1/84.6 | 73.1/63.1/62.7 | 94.4/96.0/89.0 | **97.4**/**98.2**/**93.1** |
| u block | 89.5/85.0/74.2 | 88.8/84.2/**75.5** | 62.4/48.4/51.8 | 80.1/73.9/64.3 | 75.2/68.4/67.9 | 89.7/**85.7**/75.3 | **89.9**/84.0/75.2 |
| usb | 84.9/84.3/75.1 | 78.7/79.4/69.1 | 57.0/55.3/62.9 | 87.8/88.0/78.3 | 58.9/37.4/45.7 | **92.0**/**92.2**/**84.5** | **92.0**/91.6/83.3 |
| usb adaptor | 71.1/61.4/62.2 | 76.8/71.3/64.9 | 47.5/38.4/56.5 | 80.1/74.9/**67.4** | 76.9/60.2/67.2 | 79.4/**76.0**/66.3 | **81.5**/74.5/64.9 |
| vcpill | 85.1/80.3/72.4 | 87.1/84.0/74.7 | 59.0/48.7/56.4 | 83.8/81.5/69.9 | 64.1/40.4/56.2 | 88.3/87.7/77.4 | **92.0**/**91.2**/**82.0** |
| wooden beads | 81.2/78.9/70.9 | 78.4/77.2/67.8 | 55.1/52.0/60.2 | 82.4/78.5/73.0 | 62.1/56.4/65.9 | 82.5/81.7/71.8 | **87.3**/**85.8**/**77.4** |
| woodstick | 76.9/61.2/58.1 | 80.8/72.6/63.6 | 58.2/35.6/45.2 | 80.4/69.2/60.3 | 74.1/66.0/62.1 | 80.4/69.0/63.4 | **84.0**/**73.3**/**65.6** |
| zipper | 95.3/97.2/91.2 | 98.2/98.9/95.3 | 77.2/86.7/77.6 | 96.9/98.1/93.5 | 86.0/87.0/84.0 | **99.2**/**99.6**/**96.9** | 99.1/99.5/96.5 |
| Mean | 82.4/79.0/73.9 | 83.0/80.9/74.3 | 57.2/53.4/61.5 | 82.3/79.2/73.2 | 75.6/66.4/69.9 | 86.3/84.6/77.0 | **89.3**/**86.8**/**80.2** |

Table A13: Per-class performance on **Real-IAD** dataset for multi-class anomaly localization with AUROC/AP/$F_1$-max/AUPRO metrics.

| Method → Category ↓ | RD4AD [2] CVPR'22 | UniAD [3] NeurIPS'22 | SimpleNet [13] CVPR'23 | DeSTSeg [12] CVPR'23 | DiAD [49] AAAI'24 | MambaAD [19] Arxiv'24 | Dinomaly Ours |
|---|---|---|---|---|---|---|---|
| audiojack | 96.6/12.8/22.1/79.6 | 97.6/20.0/31.0/83.7 | 74.4/0.9/4.8/38.0 | 95.5/25.4/31.9/52.6 | 91.6/1.0/3.9/63.3 | 97.7/21.6/29.5/83.9 | **98.7**/**48.1**/**54.5**/**91.7** |
| bottle cap | 99.5/18.9/29.9/95.7 | 99.5/19.4/29.6/96.0 | 85.3/2.3/5.7/45.1 | 94.5/25.3/31.1/25.3 | 94.6/4.9/11.4/73.0 | **99.7**/30.6/34.6/97.2 | **99.7**/**32.4**/**36.7**/**98.1** |
| button battery | 97.6/33.8/37.8/86.5 | 96.7/28.5/34.4/77.5 | 75.9/3.2/6.6/40.5 | 98.3/**63.9**/**60.4**/36.9 | 84.1/1.4/5.3/66.9 | 98.1/46.7/49.5/86.2 | **99.1**/46.9/56.7/**92.9** |
| end cap | 96.7/12.5/22.5/89.2 | 95.8/8.8/17.4/85.4 | 63.1/0.5/2.8/25.7 | 89.6/14.4/22.7/29.5 | 81.3/2.0/6.9/38.2 | 97.0/12.0/19.6/89.4 | **99.1**/**26.2**/**32.9**/**96.0** |
| eraser | **99.5**/30.8/36.7/96.0 | 99.3/24.4/30.9/94.1 | 80.6/2.7/7.1/42.8 | 95.8/**52.7**/**53.9**/46.7 | 91.1/7.7/15.4/67.5 | 99.2/30.2/38.3/93.7 | **99.5**/39.6/43.3/**96.4** |
| fire hood | **99.3**/30.6/37.1/**94.9** | 98.6/23.4/32.2/85.3 | 70.5/0.3/2.2/25.3 | 97.3/27.1/35.3/34.7 | 91.8/3.2/9.2/66.7 | 98.7/25.1/31.3/86.3 | **99.3**/**38.4**/**42.7**/93.0 |
| mint | 95.0/11.7/23.0/72.3 | 94.4/7.7/18.1/62.3 | 79.9/0.9/3.6/43.3 | 84.1/10.3/22.4/9.9 | 91.1/5.7/11.6/64.2 | 96.5/15.9/27.0/72.6 | **96.9**/**22.0**/**32.5**/**77.6** |
| mounts | 99.3/30.6/37.1/94.9 | **99.4**/28.0/32.8/95.2 | 80.5/2.2/6.8/46.1 | 94.2/30.0/41.3/43.3 | 84.3/0.4/1.1/48.8 | 99.2/31.4/35.4/93.5 | **99.4**/**39.9**/**44.3**/**95.6** |
| pcb | 97.5/15.8/24.3/88.3 | 97.0/18.5/28.1/81.6 | 78.0/1.4/4.3/41.3 | 97.2/37.1/40.4/48.8 | 92.0/3.7/7.4/66.5 | 99.2/46.3/50.4/93.1 | **99.3**/**55.0**/**56.3**/**95.7** |
| phone battery | 77.3/22.6/31.7/94.5 | 85.5/11.2/21.6/88.5 | 43.4/0.1/0.9/11.8 | 79.5/25.6/33.8/39.5 | 96.8/5.3/11.4/85.4 | 99.4/36.3/41.3/95.3 | **99.7**/**51.6**/**54.2**/**96.8** |
| phone battery | 77.3/22.6/31.7/94.5 | 85.5/11.2/21.6/88.5 | 43.4/0.1/0.9/11.8 | 79.5/25.6/33.8/39.5 | 96.8/5.3/11.4/85.4 | 99.4/36.3/41.3/95.3 | **99.7**/**51.6**/**54.2**/**96.8** |
| plastic nut | 98.8/21.1/29.6/91.0 | 98.4/20.6/27.1/88.9 | 77.4/0.6/3.6/41.5 | 96.5/**44.8**/**45.7**/38.4 | 81.1/0.4/3.4/38.6 | 99.4/33.1/37.3/96.1 | **99.7**/41.0/45.0/**97.4** |
| plastic plug | 99.1/20.5/28.4/94.9 | 98.6/17.4/26.1/90.3 | 78.6/0.7/1.9/38.8 | 91.9/20.1/27.3/21.0 | 92.9/8.7/15.0/66.1 | 99.0/24.2/31.7/91.5 | **99.4**/**31.7**/**37.2**/**96.4** |
| porcelain doll | 99.2/24.8/34.6/95.7 | 98.7/14.1/24.5/93.2 | 81.8/2.0/6.4/47.0 | 93.1/**35.9**/**40.3**/24.8 | 93.1/1.4/4.8/70.4 | 99.2/31.3/36.6/95.4 | **99.3**/27.9/33.9/**96.0** |
| regulator | 98.0/7.8/16.1/88.6 | 95.5/9.1/17.4/76.1 | 76.6/0.1/0.6/38.1 | 88.8/18.9/23.6/17.5 | 84.2/0.4/1.5/44.4 | 97.6/20.6/29.8/87.0 | **99.3**/**42.2**/**48.9**/**95.6** |
| rolled strip base | **99.7**/31.4/39.9/98.4 | 99.6/20.7/32.2/97.8 | 80.5/1.7/5.1/52.1 | 99.2/**48.7**/**50.1**/55.5 | 87.7/0.6/3.2/63.4 | **99.7**/37.4/42.5/**98.8** | **99.7**/41.6/45.5/98.5 |
| sim card set | 98.5/40.2/44.2/89.5 | 97.9/31.6/39.8/85.0 | 71.0/6.8/14.3/30.8 | **99.1**/**65.5**/**62.1**/73.9 | 89.9/1.7/5.8/60.4 | 98.8/51.1/50.6/89.4 | 99.0/52.1/52.9/**90.9** |
| switch | 94.4/18.9/26.6/90.9 | 98.1/33.8/40.6/90.7 | 71.7/3.7/9.3/44.2 | 97.4/57.6/55.6/44.7 | 90.5/1.4/5.3/64.2 | **98.2**/39.9/45.4/92.9 | 96.7/**62.3**/**63.6**/**95.9** |
| tape | 99.7/42.4/47.8/98.4 | 99.7/29.2/36.9/97.5 | 77.5/1.2/3.9/41.4 | 99.0/**61.7**/**57.6**/48.2 | 81.7/0.4/2.7/47.3 | **99.8**/47.1/48.2/98.0 | **99.8**/54.0/55.8/**98.8** |
| terminalblock | 99.5/27.4/35.8/97.6 | 99.2/23.1/30.5/94.4 | 87.0/0.8/3.6/54.8 | 96.6/40.6/44.1/34.8 | 75.5/0.1/1.1/38.5 | **99.8**/35.3/39.7/98.2 | **99.8**/**48.0**/**50.7**/**98.8** |
| toothbrush | 96.9/26.1/34.2/88.7 | 95.7/16.4/25.3/84.3 | 84.7/7.2/14.8/52.6 | 94.3/30.0/37.3/42.8 | 82.0/1.9/6.6/54.5 | **97.5**/27.8/36.7/**91.4** | 96.9/**38.3**/**43.9**/90.4 |
| toy | 95.2/5.1/12.8/82.3 | 93.4/4.6/12.4/70.5 | 67.7/0.1/0.4/25.0 | 86.3/8.1/15.9/16.4 | 82.1/1.1/4.2/50.3 | **96.0**/16.4/25.8/86.3 | 94.9/**22.5**/**32.1**/**91.0** |
| toy brick | 96.4/16.0/24.6/75.3 | **97.4**/17.1/27.6/**81.3** | 86.5/5.2/11.1/56.3 | 94.7/24.6/30.8/45.5 | 93.5/3.1/8.1/66.4 | 96.6/18.0/25.8/74.7 | 96.8/**27.9**/**34.0**/76.6 |
| transistor1 | 99.1/29.6/35.5/95.1 | 98.9/25.6/33.2/94.3 | 71.7/5.1/11.3/35.3 | 97.3/43.8/44.5/45.4 | 88.6/7.2/15.3/58.1 | 99.4/39.4/40.0/96.5 | **99.6**/**55.3**/**53.3**/**97.8** |
| u block | **99.6**/40.5/45.2/**96.9** | 99.3/22.3/29.6/94.3 | 76.2/4.8/12.2/34.0 | 96.9/**57.1**/**55.7**/38.5 | 88.8/1.6/5.4/54.2 | 99.5/37.8/46.1/95.4 | 99.5/41.8/45.6/96.8 |
| usb | 98.1/26.4/35.2/91.0 | 97.9/20.6/31.7/85.3 | 81.1/1.5/4.9/52.4 | 98.4/42.2/47.7/57.1 | 78.0/1.0/3.1/28.0 | **99.2**/39.1/44.4/95.2 | **99.2**/**45.0**/**48.7**/**97.5** |
| usb adaptor | 94.5/9.8/17.9/73.1 | 96.6/10.5/19.0/78.4 | 67.9/0.2/1.3/28.9 | 94.9/**25.5**/**34.9**/36.4 | 94.0/2.3/6.6/75.5 | 97.3/15.3/22.6/82.5 | **98.7**/23.7/32.7/**91.0** |
| vcpill | 98.3/43.1/48.6/88.7 | **99.1**/40.7/43.0/91.3 | 68.2/1.1/3.3/22.0 | 97.1/64.7/62.3/42.3 | 90.2/1.3/5.2/60.8 | 98.7/50.2/54.5/89.3 | **99.1**/**66.4**/**66.7**/**93.7** |
| wooden beads | 98.0/27.1/34.7/85.7 | 97.6/16.5/23.6/84.6 | 68.1/2.4/6.0/28.3 | 94.7/38.9/42.9/39.4 | 85.0/1.1/4.7/45.6 | 98.0/32.6/39.8/84.5 | **99.1**/**45.8**/**50.1**/**90.5** |
| woodstick | 97.8/30.7/38.4/85.0 | 94.0/36.2/44.3/77.2 | 76.1/1.4/6.0/32.0 | 97.9/**60.3**/**60.0**/51.0 | 90.9/2.6/8.0/60.7 | 97.7/40.1/44.9/82.7 | **99.0**/50.9/52.1/**90.4** |
| zipper | 99.1/44.7/50.2/96.3 | 98.4/32.5/36.1/95.1 | 89.9/23.3/31.2/55.5 | 98.2/35.3/39.0/78.5 | 90.2/12.5/18.8/53.5 | **99.3**/58.2/61.3/97.6 | **99.3**/**67.2**/**66.5**/**97.8** |
| Mean | 97.3/25.0/32.7/89.6 | 97.3/21.1/29.2/86.7 | 75.7/2.8/6.5/39.0 | 94.6/37.9/41.7/40.6 | 88.0/2.9/7.1/58.1 | 98.5/33.0/38.7/90.5 | **98.8**/**42.8**/**47.1**/**93.9** |

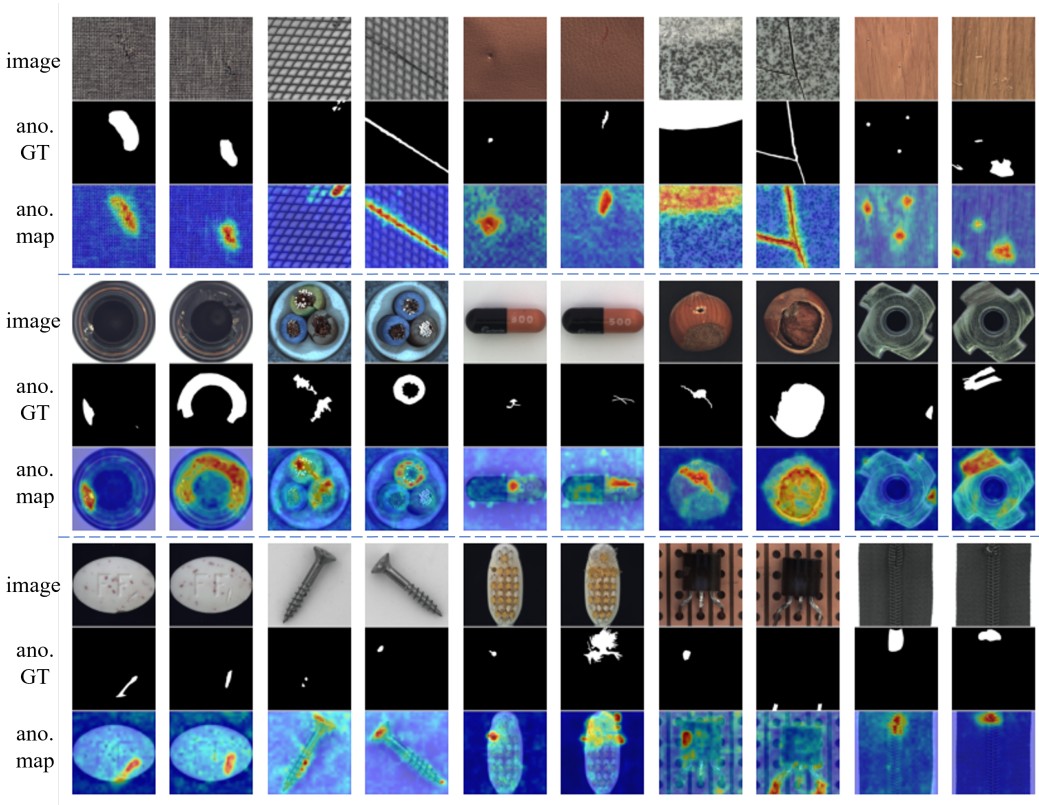

Figure A1: Anomaly maps visualization on MVTec-AD. All samples are randomly chosen.

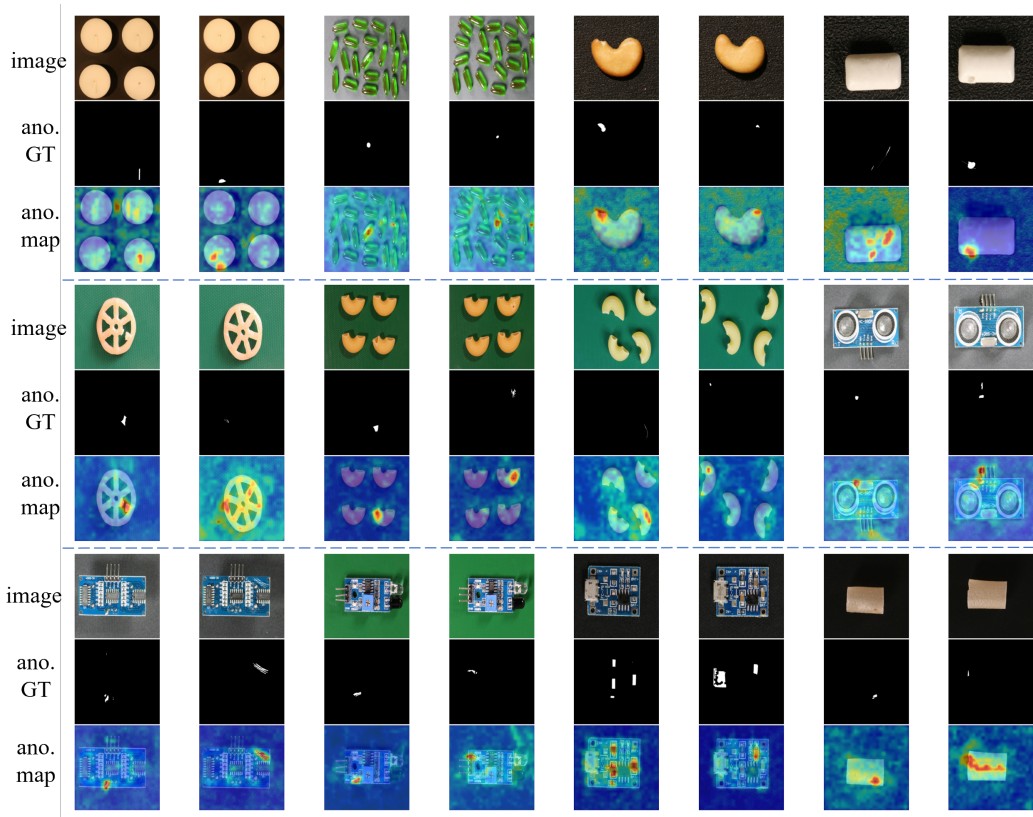

Figure A2: Anomaly maps visualization on VisA. All samples are randomly chosen.

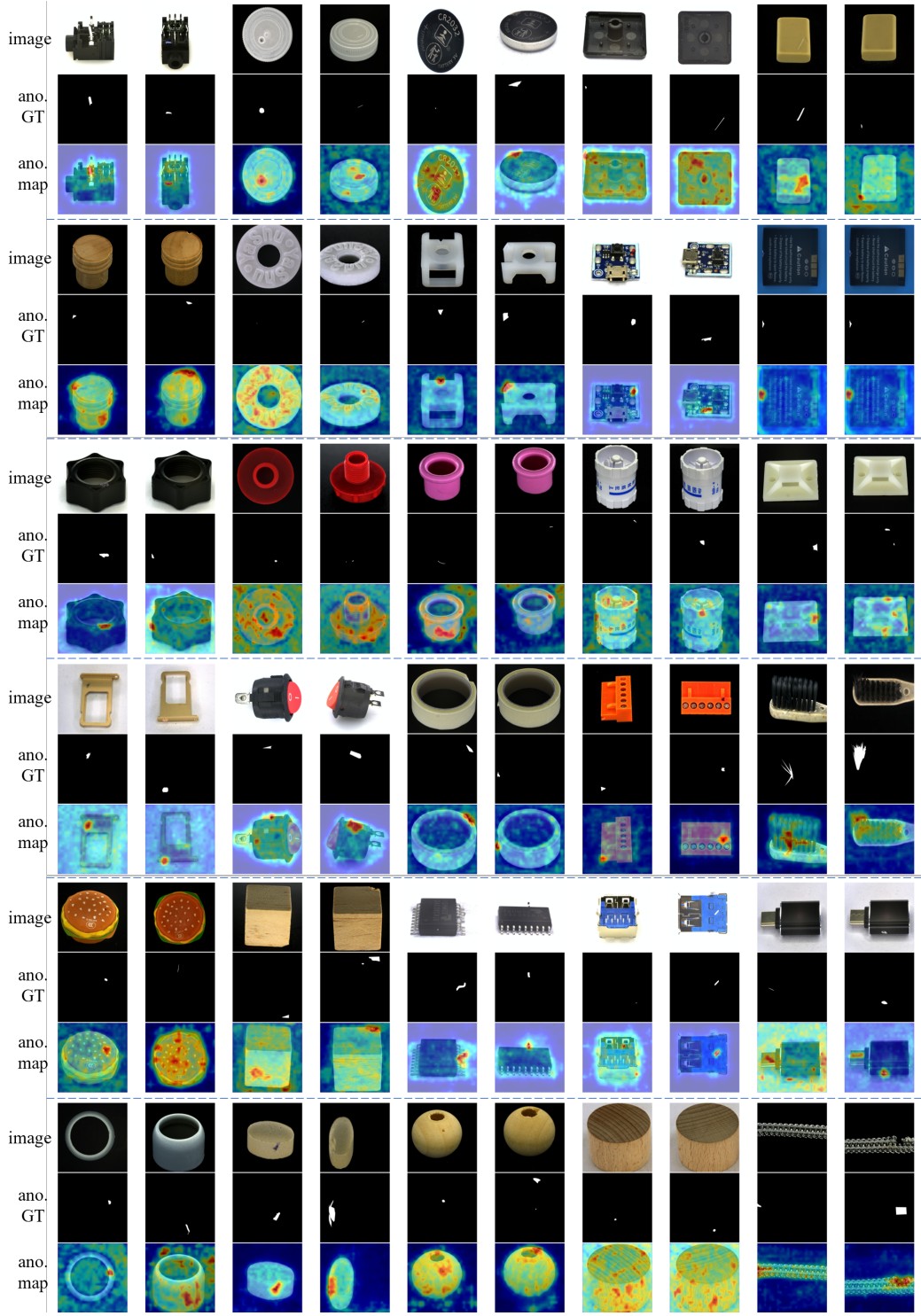

Figure A3: Anomaly maps visualization on Real-IAD. All samples are randomly chosen.

