# OpenReview forum: "Dinomaly: The Less Is More Philosophy in Multi-Class Unsupervised Anomaly Detection"
_NeurIPS.cc/2024/Conference — Submitted to NeurIPS 2024_

### Official Review · Reviewer_m7bP · 2024-07-09

**Soundness:** 3
**Presentation:** 2
**Contribution:** 3
**Rating:** 6
**Confidence:** 4

**Summary:**

The paper "Dinomaly: The Less Is More Philosophy in Multi-Class Unsupervised Anomaly Detection" introduces a minimalist reconstruction-based framework for unsupervised anomaly detection (UAD) in multi-class settings. The framework focuses on four main components: Foundation Transformers, Noisy Bottleneck, Linear Attention, and Loose Reconstruction. Extensive experiments on MVTec-AD, VisA, and Real-IAD datasets show that Dinomaly achieves superior performance compared to state-of-the-art multi-class and even some class-separated UAD methods.

**Strengths:**

1. Using simple components like Foundation Transformers, Noisy Bottleneck, Linear Attention, and Loose Reconstruction to achieve superior performance is highly original. This is a significant departure from traditional methods that rely on complex designs and multiple modules. It challenges the conventional views: more complex architectures are necessary for better performance in anomaly detection tasks.
2. The methodology is well-detailed, and the experimental design is robust. The authors conduct extensive experiments on three well-known datasets (MVTec-AD, VisA, and Real-IAD), providing comprehensive performance metrics and comparisons with SOTA methods. The result is convincing, showing that Dinomaly not only outperforms existing MUAD methods but also surpasses some of the best class-separated UAD methods.
3. The paper is generally clear, well-organized, and relatively reproducible.
4. The significance of this work is substantial, and makes a valuable contribution to anomaly detection, as it addresses a major challenge in UAD—achieving high performance in multi-class settings without resorting to complex, specialized architectures, and is potentially scalable.

**Weaknesses:**

1. The paper provides a detailed explanation of the proposed framework but lacks important justification and discussion, it is difficult for readers to realize the novelty and improvements brought by Dinomaly. The author may need to compare Dinomaly to specific previous methods, highlighting the differences and improvements. Discuss how the minimalist approach contrasts with more complex architectures and why this improvement is significant.
2. The motivations for choosing Noisy Bottleneck and Loose Reconstruction are not deeply explored. For instance, explain in more detail why Noisy Bottleneck helps prevent identity mapping.
3. The paper claims simplicity but there was no discussion of parameter number, computational complexity, or time complexity in the experiment.
4. In Loose Constraint, the author claims that 1-group LC mixes low-level and high-level features which is harmful for anomaly localization. How to group the features into the low-semantic-level group and high-semantic-level group in 2-group LC?

**Questions:**

1. Can you provide more detailed justifications for the choice of Noisy Bottleneck and Loose Reconstruction? Why do these help in preventing identity mapping and improving anomaly detection performance?
Suggestion: A detailed theoretical explanation or additional references would clarify the rationale of the design of Dinomaly and strengthen the argument for its effectiveness.
2. Provide a more explicit comparison to recent methods in terms of performance and conceptual differences. Highlight any specific limitations of prior work that Dinomaly addresses.
3. Increase discussion on simplicity and scalability.
4. The credibility of the paper may benefit from deeper experimental analysis. Known that Dinomaly surpasses compared methods by a large margin on all datasets and all metrics, if its performance is limited under certain conditions, such as when the input is video?

---

> ### Author Rebuttal · Authors · 2024-08-05
>
> First, thank you for your valuable reviews and comments.
>
> __W1,2&Q1,2: Justification and discussion of proposed components, and comparison of the difference with prior works.__
>
> In Noisy Bottleneck, we show that the simple Dropout can work as a noise injection module to transformer "reconstruction" to "restoration".  The base motivation and insight is that when a decoder is trained to reconstruct its input (output==input), it can generalize so well that it can reconstruct unseen samples (so-called identity mapping) because there is no explicit training regularization that forbids the decoder from generating unseen input. However, if the decoder is trained to restore (a.k.a. denoise) the input given noisy input (output != input), there is theoretically no over-generalization problem because there is an explicit training goal that forces the decoder to generate normal feature given abnormal input.
>
> In the original Dropout thesis [a] (page 2): "Dropout can also be interpreted as a way of regularizing a neural network by adding noise to its hidden units. This idea has previously been used in the context of Denoising Autoencoders [b] [c] where noise is added to the inputs of an autoencoder and the target is kept noise-free." This "ancient" Denoising Auto-Encoder is very similar to the denoising paradigm of Dinomaly. Therefore, there is theoretical evidence to adopt Dropout as noise injection.
>
> We have primarily discussed the limitations of previous noise injection methods in L115-L117, that their anomaly generation strategies are heuristic and hand-crafted which are not universal across domains, datasets, and methods. This leads to the advantage of our Dropout-based Noisy Bottleneck-----simplicity. Furthermore, we compare Dropout-based Noisy Bottleneck with Feature Jitting proposed in UniAD in Appendix Table A6, which demonstrates our effectiveness and robustness.
>
> For Loose Reconstruction, we connect feature reconstruction to feature-level knowledge distillation in L160-L158. To be more specific, in prior works on feature-level knowledge distillation[d][e], it is suggested that dense layer-wise distillation helps the student model to better mimic the knowledge of the teacher and results in better generalization, which is harmful in UAD context.
>
> We will include and shorten the above discussion and information in the final version.
>
> [a] Srivastava, Nitish. Improving neural networks with dropout. University of Toronto
>
> [b] Extracting and composing robust features with denoising autoencoders.  ICML ’08
>
> [c] Stacked denoising autoencoders: Learning useful representations in a deep network with a local denoising criterion. Journal of Machine Learning Research, 2010
>
> [d]Patient knowledge distillation for bert model compression." arXiv preprint arXiv:1908.09355 (2019)
>
> [e]Task-aware layer-wise distillation for language model compression." International Conference on Machine Learning. PMLR, 2023.
>
>
> __W3&Q3: Parameter number, computational complexity, or time complexity. Increase discussion on simplicity and scalability.__
>
> The computation cost of Dinomaly was presented in Table A2 in the Appendix, including parameters (148M), MACs (104.7G), and latency per image (17.8 ms) for the default ViT-B version. We will additionally include the total training time (about 1.5 hours on one NVIDIA 3090) in the final version.
>
> The scalability is presented in Table A3 in the Appendix, where Dinomaly is scaled from ViT-Small to ViT-Large. Results show that our method follows the "scaling law".
>
> The above information will be moved to the main paper given more pages in the final version.
>
> __W4: How to group the features into the low-semantic-level group and high-semantic-level group in 2-group LC?__
>
> As shown in Figure 2, the lower four layers are grouped as low-semantic-level; the deeper four layers are grouped as high-semantic-level. This scheme is simple, following the common sense of neural networks that shallow layers extract basic visual features such as lines, colors, borders, and corners, while deep layers extract abstract semantic information.
>
> __Q4: Performance under other conditions, such as video. When is Dinomaly limited?__
>
> The proposed method is not directly applicable to video modality, because video anomaly detection methods usually adopt temporal-spatial networks. As a complement, we include datasets of three more image domains, including MPDD (metal parts, position not aligned), BTAD (noisy training set), and Uni-Medical (medical images, including brain MRI, liver CT, and retinal OCT). A preliminary comparison is shown as below. As presented in the tables, the superiority of Dinomaly is relatively limited under noisy training set (anomaly images in normal training set) and medical domains.
>
> The full results will be included in Appendix as further information on a wide range of domains.
>
> Short comparison on MPDD.
> | Method | I-AUROC | I-AP | P-AUROC | P-AP |
> |---|---|---|---|---|
> | RD4AD | 90.3 | 92.8 | 98.3 | 39.6 |
> | UniAD | 80.1 | 83.2 | 95.4 | 19.0 |
> | ViTAD | 87.4 | 90.8 | 97.8 | 34.6 |
> | Dinomaly | 97.2 | 98.4 | 99.1 | 59.5 |
>
> Short comparison on BTAD.
> | Method | I-AUROC | I-AP | P-AUROC | P-AP |
> |---|---|---|---|---|
> | RD4AD | 94.1 | 96.8 | 98.0 | 57.1 |
> | UniAD | 94.5 | 98.4 | 97.4 | 52.4 |
> | ViTAD | 94.0 | 97.0 | 97.6 | 58.3 |
> | Dinomaly | 95.4 | 98.4 | 97.8 | 70.1 |
>
> Short comparison on Uni-Medical.
> | Method | I-AUROC | I-AP | P-AUROC | P-AP |
> |---|---|---|---|---|
> | RD4AD | 76.4 | 75.8 | 96.4 | 38.9 |
> | UniAD | 80.4 | 76.6 | 96.5 | 39.0 |
> | ViTAD | 81.5 | 80.6 | 97.0 | 46.8 |
> | Dinomaly | 83.4 | 82.7 | 96.7 | 50.9 |
>
> [a] Deep learning-based defect detection of metal parts: evaluating current methods in complex conditions. In ICUMT, 2021
>
> [b]Vt-adl: A vision transformer network for image anomaly detection and localization. In ISIE, 2021.
>
> [c] ADer: A Comprehensive Benchmark for Multi-class Visual Anomaly Detection. arXiv:2406.03262 2024.

---

> > ### Comment · Reviewer_m7bP · 2024-08-12
> >
> > I agree with the author's answer to “Why choose Noisy Bottleneck and differentiation”, but still do not explain why dense layer-wise distillation is harmful to UAD.
> > Thanks for the valuable comments by reviewer #LXBB. I agree with the statement that “it is challenging to include the suggested experiments in the main text without allowing major changes in subsequent submissions”. There are indeed some writing shortcomings in innovative ideas, motivations, clarity of principles, and related work. But in terms of technical contribution, I acknowledge the good performance and comprehensive experimental design of this work, as the author replied, “We provide a variety of selections for downstream users to choose from according to their budget”, so I will keep my decision.

---

> ### Author Response · Authors · 2024-08-12
> **Response**
>
> Thank you for your valuable review and comments.
>
> As discussed in the 4th paragraph of the rebuttal of W1,2&Q1,2, we have attributed the harms of dense layer-wise distillation to its generalization ability.  Apart from the listed [d] and [e], there were plenty of works that utilized dense layer-wise distillation schemes for better generalization in the context of knowledge distillation. E.g., In [f]: "Given that the teacher’s layerwise representations often contain rich semantic knowledge, they can significantly improve the student’s generalizability". Because we have established that over-generalization is a curse in the context of UAD, such dense layer-wise distillation paradigm is self-evidently harmful to reconstruction/distillation-based UAD. In addition, as discussed in L165-L166, "the student (decoder) can better mimic the behavior of the teacher (encoder) given more layer-to-layer supervision", which is also clearly unwanted in UAD, because an "Identical" student cannot detect anomalies based on the reconstruction error. We believe such analysis can be a strong conceptual principle for the proposed Loose Constraint.
>
> [f] Liang, Chen, et al. "Module-wise adaptive distillation for multimodality foundation models." Advances in Neural Information Processing Systems 36 (2023).
>
> Due to the page limit, we have to put extensive ablation studies in the Appendix, while leaving the main paper for presenting contributions.  We will present at least two more ablation experiments in the final version, while others can be included in Appendix with proper references.
>
> Again, thank you for your thorough review. Looking forward to further discussion.

---

### Official Review · Reviewer_hJXx · 2024-07-12

**Soundness:** 3
**Presentation:** 4
**Contribution:** 3
**Rating:** 6
**Confidence:** 4

**Summary:**

This paper focuses on the Multi-class Unsupervised Anomaly Detection task and proposes a minimalistic reconstruction-based anomaly detection framework — Dinomaly that consists of only vanilla Transformer blocks. In this framework, four key components (Foundation Transformers, Noisy Bottleneck, Linear Attention, and Loose Reconstruction) are introduced to alleviate the performance gap between multi-class and class-separated models. The paper conducts extensive experiments on three major datasets: MVTec-AD, VisA, and Real-IAD. Results show that Dinomaly outperforms current state-of-the-art methods.

**Strengths:**

(1)The paper is well-written and has clear statements which make it easy to understand.

(2)The design of Dinomaly is straightforward but innovative. The use of foundation transformers, noisy bottleneck, linear attention, and loose reconstruction is well-justified.

(3)The paper generally outperformed existing SOTA methods and did enough experiments and comparisons.

**Weaknesses:**

(1)The method relies heavily on transformer architectures, which might limit its applicability to other types of models.

(2)Transformers can be resource-intensive, and the paper does not fully address the computational cost of training and inference.

(3)The method's generalization to other domains or types of anomaly detection is not fully explored.

**Questions:**

(1) Please specify the computational overhead of your work.

(2) There have been various recent works that attempt to perform anomaly detection in a more general zero-shot/few-shot setting, where the model trained on multiple classes is used to test samples from unseen classes. A discussion of these recent works and how the zero-shot (cross-dataset) performance of this approach should be added. e.g.,

"WinCLIP: Zero-/few-shot anomaly classification and segmentation" CVPR'2023

"AnomalyCLIP: Object-agnostic Prompt Learning for Zero-shot Anomaly Detection" ICLR'2024

"Toward Generalist Anomaly Detection via In-context Residual Learning with Few-shot Sample Prompts" CVPR'2024

"PromptAD: Learning Prompts with only Normal Samples for Few-Shot Anomaly Detection" CVPR'2024

(3) If I understand correctly, the best and second-best results in each table should be highlighted, with the best results in bold and the second-best results underlined. However, in Table 2, the best results for MVTec-AD (P-AP) and VisA (P-AP) are incorrectly marked. Additionally, could you clarify why the Dinomaly (MUAD) model outperforms the Dinomaly (class-separated) model in this metric?

**Limitations:**

Limitations are discussed in Supplementary Sec. A.4.

---

> ### Author Rebuttal · Authors · 2024-08-05
>
> Thank you for your valuable reviews and comments.
>
> __W1: The method relies heavily on transformer architectures, which might limit its applicability to other types of models.__
>
> Transformer architecture has proven its ability as the foundation of machine learning tasks, including NLP (GPT, Llama) and CV (DINO, SAM). We believe that it is important to fully harness the power of foundational Transformers in UAD context, though some of the discoveries are Transformer-specialized. In addition, the insights of proposed components can be adapted to other architectures with proper modification.
>
> __W2&Q1:Transformers can be resource-intensive, and the paper does not fully address the computational cost of training and inference. Please specify the computational overhead.__
>
> The computation cost of Dinomaly was presented in Table A2 in the Appendix, including parameters (148M), MACs (104.7G), and latency per image (17.8 ms). The total training computation can be inferred by total iterations and MACs per image. We will additionally include the total training time (about 1.5 hours on one NVIDIA 3090) in the final version. The latency per image of Dinomaly-ViT-Base is 17.8ms (~58FPS) on a single NVIDIA RTX 3090 (consumer-level GPU), which can be considered enough for industrial applications.
>
> __W3: The method's generalization to other domains or types of anomaly detection is not fully explored.__
>
> We conduct experiments on more public datasets that represent other domains, including MPDD [a] (metal parts, position not aligned), BTAD [b] (noisy dataset), and Uni-Medical [c] (medical images, including brain MRI, liver CT, and retinal OCT). A preliminary comparison is shown as below. The full results will be included in Appendix as further information on a wide range of domains.
>
> Short comparison on MPDD.
> | Method | I-AUROC | I-AP | P-AUROC | P-AP |
> |---|---|---|---|---|
> | RD4AD | 90.3 | 92.8 | 98.3 | 39.6 |
> | UniAD | 80.1 | 83.2 | 95.4 | 19.0 |
> | ViTAD | 87.4 | 90.8 | 97.8 | 34.6 |
> | Dinomaly | 97.2 | 98.4 | 99.1 | 59.5 |
>
> Short comparison on BTAD.
> | Method | I-AUROC | I-AP | P-AUROC | P-AP |
> |---|---|---|---|---|
> | RD4AD | 94.1 | 96.8 | 98.0 | 57.1 |
> | UniAD | 94.5 | 98.4 | 97.4 | 52.4 |
> | ViTAD | 94.0 | 97.0 | 97.6 | 58.3 |
> | Dinomaly | 95.4 | 98.4 | 97.8 | 70.1 |
>
> Short comparison on Uni-Medical.
> | Method | I-AUROC | I-AP | P-AUROC | P-AP |
> |---|---|---|---|---|
> | RD4AD | 76.4 | 75.8 | 96.4 | 38.9 |
> | UniAD | 80.4 | 76.6 | 96.5 | 39.0 |
> | ViTAD | 81.5 | 80.6 | 97.0 | 46.8 |
> | Dinomaly | 83.4 | 82.7 | 96.7 | 50.9 |
>
> [a] Deep learning-based defect detection of metal parts: evaluating current methods in complex conditions. In ICUMT, 2021
>
> [b]Vt-adl: A vision transformer network for image anomaly detection and localization. In ISIE, 2021.
>
> [c] ADer: A Comprehensive Benchmark for Multi-class Visual Anomaly Detection. arXiv preprint arXiv:2406.03262 2024.
>
> __Q2: Recent works on few-shot/zero-shot anomaly detection.__
>
> Thanks for the valuable suggestion. Few-shot/zero-shot UAD is a new promising UAD setting proposed and studied lately. We also have been devoted to this field. We will discuss recent few-shot/zero-shot works in the Related Work in the final version. Nevertheless, this setting is a very different track from MUAD. Dinomaly is not capable of adapting to few/zero-shot/cross-dataset settings. Vice versa, few/zero-shot methods lag far behind the proposed Dinomaly and other MUAD works in absolute detection performance. We will include this information in the Limitation section.
>
> __Q3-1:  In Table 2, the best results for MVTec-AD (P-AP) and VisA (P-AP) are incorrectly marked.__
>
> In Table 2, we intend to present Dinomaly's performance under the conventional class-separated UAD setting. Methods trained under class-separated setting (starting from the 2nd row) are compared together, leaving Dinomaly (MUAD) out as a reference. Therefore, Dinomaly (MUAD) is not in bold or underlined, but in italic.
>
> __Q3-2: Why does the Dinomaly (MUAD) model outperform the Dinomaly (class-separated) in AP on MVTec-AD and VisA?__
>
> Dinomaly localizes and segments anomalous regions by reconstruction error. In class-separated setting with few images, the decoder of Dinomaly is under-optimized so that the decoder is too sensitive to non-anomaly local deviation. This phenomenon does not affect image-level performance because such reconstruction error does not become the largest error in an image, but it affects pixel AUC metrics (AP and AUROC) as they measure the performance integration across the whole anomaly score range.

---

> > ### Comment · Reviewer_hJXx · 2024-08-13
> >
> > Thanks for the authors' response, and it basically addressed my concerns. I will maintain my score.

---

### Official Review · Reviewer_LXBB · 2024-07-12

**Soundness:** 3
**Presentation:** 3
**Contribution:** 3
**Rating:** 5
**Confidence:** 5

**Summary:**

This paper introduces Dinomaly, a simple yet effective anomaly detection framework using pure Transformer architectures. It identifies four key components essential for multi-class anomaly detection: Foundation Transformers, Noisy Bottleneck, Linear Attention, and Loose Reconstruction. Extensive experiments on MVTec-AD, VisA, and Real-IAD datasets show that Dinomaly achieves superior performance, surpassing both state-of-the-art multi-class and class-separated anomaly detection methods.

**Strengths:**

1. The authors have conducted extensive experiments to validate the effectiveness of their method across multiple anomaly detection tasks.
2. The authors have proposed a simple yet effective framework that approaches and even surpasses the results of state-of-the-art methods in single-class anomaly detection tasks.

**Weaknesses:**

1. L53-55 'In addition, previous...': The authors should provide relevant evidence rather than subjective assumptions.
2. L77: Placing the Related Works section in the appendix is unconventional.
3. The first component proposed by the authors, Foundation Transformers, was already introduced in the ViTAD paper, which diminishes the overall contribution of the paper.
4. The input resolution used in the authors' experiments is 448x448, while other comparison methods use 256x256 or 224x224. This is an extremely unfair comparison. Please include results with a 256 resolution in table for a fair comparison.
5. In the proposed Loose Loss, 90% of the feature points were selected. How was this 90% hyperparameter determined? Please provide ablation study results.
6. The authors have employed Linear Attention to reduce computational load while maintaining similar performance. It is recommended that the authors compare the parameter count and FLOPs of their method with those of the baseline methods to demonstrate its efficiency. Additionally, it is suggested to conduct ablation studies to verify the computational efficiency of Linear Attention.
7. Other methods, such as RD4AD, SimpleNet, and UniAD, perform under the proposed settings. The authors can conduct a more equitable comparison.

**Questions:**

See the Weaknesses.

=== After Rebuttal ===
I decide to raise my score from 4 to 5.

**Limitations:**

It is recommended that the authors evaluate the performance of different methods under fair settings.

---

> ### Author Rebuttal · Authors · 2024-08-04
>
> First, thank you for your valuable reviews and comments.
>
> __W1:  Relevant evidence rather than subjective assumptions for L53-55.__
>
> Previous works on MUAD do make large efforts to design special modules for mitigating "identity mapping". Taking works of NeurIPS as examples, UniAD (NIPS22), the pioneer of MUAD, hacks into the self-attention mechanism by masking the neighbor's attention and designs complex decoder architecture that is distinct from vanilla ViT. HVQ-Trans (NIPS23), taking steps further, maintains a prototype set (memory bank) for each decoder layer which requires extra hyperparameter tuning for the code book and its POT loss. ReContrast (NIPS23) employs two encoders, one frozen and one trainable, for cross-contrastive-reconstruction, which largely alters the naive reconstruction paradigm.
>
> Such methods are considered to be rather complicated, leaving challenges for downstream users to understand and tune on their datasets. In addition, complicated design restricts the generalization of method; e.g., UniAD developed for MVTec-AD performs worse than the more simple RD4AD when used on VisA (88.8 vs. 92.4). These lead to the advantage of our proposed simple and concise Dinomaly. We will include the above discussion in the final version.
>
>
> __W2: Placing the Related Works section in the appendix is unconventional.__
>
> Yes, indeed, due to the limited pages in the submission. We will place it in the main content given one more page in the final version.
>
> __W3: Foundation Transformers, was already introduced in the ViTAD paper.__
>
> The use of foundation Transformers is a core component of Dinomaly from a narrative perspective; but, we consider it as an important component, but not the main novelty. That is why it is not included in the component ablation study in Table3. However, we do want to emphasize the importance of a powerful foundation model to readers. Hence, we include it as a core element of Dinomaly for the integrity of writing. For the first time, we extensively present the choice of pre-trained foundations (Table A4) (DeiT, MAE, DINO, iBOT, BEiT... etc.). We hope such exploration will inspire and help future research to select proper backbones in UAD tasks.
>
> __W4: Unfair comparison to other methods with 256/224 input.__
>
> The comparison is based on the best performance, because 256x256 is already the optimal resolution for the compared methods. In our reproduction experiments, prior methods got worse performance when increasing input size from their default 256x256, as shown in this Table. Therefore, our comparison is based on the optimum vs. optimum. We will add the above explanation in the final version.
>
> In addition, the results of Dinomaly with different input sizes are presented in Table A3/A4, where the performance of low resolution still exceeds previous SoTAs by a large margin.
>
> | Method | Input size | I-AUROC | P-AUROC |
> |---|---|---|---|
> | RD4AD | __256x256__(best) | 94.6 | 96.1 |
> | RD4AD | 320x320 | 93.2 | 95.7 |
> | RD4AD | 384x384 | 91.9 | 94.9 |
> | ViTAD |__256x256__(best)  | 98.3 | 97.7 |
> | ViTAD | 320x320 | 98.3 | 97.6 |
> | ViTAD | 384x384 | 97.8 | 97.5 |
> | ReContrast | __256x256__(best)  | 98.3 | 97.1 |
> | ReContrast | 320x320 | 98.2 | 96.8 |
> | ReContrast | 384x384 | 95.2 | 96.5 |
>
> __W5: The selection of hyperparameter in Loose Loss.__
>
> The discarding rate is extremely robust, as shown in the following Table. We will include this ablation in the Appendix of the final version. We follow a simple intuition to discard the majority of easy samples by setting a round number 90% without tuning. Tuning this hyperparameter can result in even better results.
>
> | Discard rate | I-AUROC | P-AUROC |
> |---|---|---|
> | 95% | 99.58 | 98.32 |
> | 90% (Default) | 99.60 | 98.35 |
> | 80% | 99.65 | 98.38 |
> | 70% | 99.64 | 98.37 |
> | 60% | 99.63 | 98.34 |
>
> __W6: Computation cost of Linear Attention__
>
> We have presented the computation cost (parameters, MACs(~0.5*FLOPs), latency) of Dinomaly model in Table A2. We will add the computation cost of other methods as a comparison in the final version. It also shows that Dinomaly is scalable.
>
> In addition, the MACs of Softmax Attention and Linear Attention are 2.82G and 1.86G (each attn layer). Their parameters are exactly the same (2.36M). The cost of other modules, e.g. encoder and MLP, are not affected.
>
> The adoption of Linear Attention is driven by its "unfocus" ability which can alleviate identity mapping and improve UAD performance. Currently，the reduction of computation is not the main concern of this paper.
>
> | Method | Parameters | FLOPs | I-AUROC | P-AUROC |
> |---|---|---|---|---|
> | RD4AD | 80.6M | 28.4G | 94.6 | 96.1 |
> | SimpleNet | 72.8M | 17.2G | 95.3 | 96.9 |
> | ViTAD | 39.0M | 9.7G | 98.3 | 97.7 |
> | DiAD | 1331M | 451.5G | 97.2 | 96.8 |
> | Dinomaly-Small-384x384 | 37.4M | 60.2G | 99.3 | 98.1 |
> | Dinomaly-Base-280x280 | 148M | 111.5G | 99.5 | 98.4 |
> | Dinomaly-Base-384x384 (default) | 148M | 210.1G | 99.6 | 98.4 |
> | Dinomaly-Base-Softmax-384x384 | 148M | 224.7G | 99.5 | 98.2 |
> | Dinomaly-Large-384x384 | 413.5M | 571.0G | 99.8 | 98.5 |
>
> __W7: Other methods, such as RD4AD, SimpleNet, and UniAD, perform under the proposed settings. The authors can conduct a more equitable comparison.__
>
> All methods in Table 1 are under the setting of one-model multi-class setting, which is the setting of our proposed method.
>
>  If you mean "setting" by the same input size, as the response of Weakness 4, the original resolution of the compared method is nearly the optimal resolution. UAD is different from classical CV tasks like ImageNet classification where the higher resolution must result in better performance.  Therefore, it is a fair comparison of optimum vs. optimum. It is less reasonable to force the same input size for all methods as different methods and encoders have different natures; e.g., ResNet with 256x256 input can utilize 64x64 feature maps, while ViT with 256x256 input has only 16x16 feature maps.

---

> > ### Author Response · Authors · 2024-08-07
> > **Revision of the Last Table of Rebuttal**
> >
> > In the last table of our rebuttal, the computation cost (FLOPs) of compared methods (RD4AD, SimpleNet, ViTAD, DiAD) is drawn from the benchmark paper ADer [a]. However, we just found that they confused MACs(multiply–accumulate operations) with FLOPs(floating point operations). MACs~0.5*FLOPs. The FLOPs they reported are acutally MACs, which are around half of the real FLOPs.
> >
> > Therefore, we report GMACs for unification.
> >
> > [a] ADer: A Comprehensive Benchmark for Multi-class Visual Anomaly Detection. arXiv preprint arXiv:2406.03262 2024.
> >
> > | Method | Parameters | MACs | I-AUROC | P-AUROC |
> > |---|---|---|---|---|
> > | RD4AD | 80.6M | 28.4G | 94.6 | 96.1 |
> > | SimpleNet | 72.8M | 17.2G | 95.3 | 96.9 |
> > | DRAEM | 97.4M | 198G | 54.5 | 47.6 |
> > | ViTAD | 39.0M | 9.7G | 98.3 | 97.7 |
> > | DiAD | 1331M | 451.5G | 97.2 | 96.8 |
> > | Dinomaly/ViT-Small-384x384 | 37.4M | 26.2G | 99.3 | 98.1 |
> > | Dinomaly/ViT-Base-280x280 | 148M | 53.7G | 99.5 | 98.4 |
> > | Dinomaly/ViT-Base-384x384 (default) | 148M | 104.6G | 99.6 | 98.4 |
> > | Dinomaly/ViT-Base-Softmax-384x384 | 148M | 112.3G | 99.5 | 98.2 |
> > | Dinomaly/ViT-Large-384x384 | 413.5M | 285.3G | 99.8 | 98.5 |

---

> > ### Comment · Reviewer_LXBB · 2024-08-10
> > **For Authors**
> >
> > The authors have resolved a few issues, but most concerns remain unaddressed or have been evaded:
> >
> > Q1. Placing related works in the appendix violates the formatting guidelines, potentially extending the main text to 10 pages instead of 9, which is unfair to other works. Considering that revised papers generally increase in length, I believe it will be challenging for the authors to include the Related Work section in the main text as required.
> >
> > Q3. The authors list ViT as a contribution point in lines L57-59 and Section 2.1, particularly the use of DINOv2 weights, which offers no technical contribution to AD (as also noted by Reviewer **#m7bP**). However, I believe this is a core aspect of the work. The authors should apply other modules to different frameworks to validate the effectiveness of their contributions.
> >
> > Q4/Q6. 1) A comparison of model efficiency at the standard resolution of 256 is necessary to ensure a fair comparison of different methods under the same parameter and computation constraints. Please provide the comparison results. 2) For perception tasks, pixel-level metrics should increase with resolution, but the results provided by the authors show the opposite trend. Please explain this. 3) AUROC is an unreliable metric; providing results with additional metrics would be beneficial. 4) Notably, the model performance under several weights in Table A4, such as MAE, significantly deteriorates, weakening the evidence for the effectiveness of other contributions. How does the performance compare when using stronger pre-trained weights for the comparison methods? 5) The reviewers do not accept the authors' claim of focusing solely on model performance without considering computational cost, which is unreasonable for model applications. This issue was also raised by reviewers **#hJXx** and **#m7bP**. The authors should provide a comparison of the proposed method scaled to a similar level as the comparison methods.

---

> ### Author Response · Authors · 2024-08-11
> **Response (1/2)**
>
> Response(1/2)
>
> Thank you for your valuable comments. We by no means intended to evade any concerns.  We spare no effort to clarify any raised concerns.
>
> __R-Q1:__
>
> NeurIPS allows one more page for the final version (10 pages), which will fit a condensed Related Work.
>
> In addition, though there is no explicit Related Work section in our submission, we merged such content directly in Introduction (L28-L55). We discussed categories of conventional UAD methods, why conventional methods fail in MUAD, and recent methods of MUAD, which elaborates the background and history of MUAD. Such narrative will also be partly moved to the final Related Work.
>
> Moreover, we have checked the NeurIPS formatting guidelines carefully and found no such guideline that strictly demands an explicit Related Work section. Many NeurIPS papers do not have an explicit Related Work section, some not even in the final version.
>
> To name a few, see:
>
> https://openreview.net/pdf?id=DP2lioYIYl
>
> https://openreview.net/pdf?id=aExAsh1UHZo
>
> https://openreview.net/forum?id=hgLMht2Z3L
>
> https://openreview.net/forum?id=wxkBdtDbmH&noteId=JKLz5pP5sJ
>
> https://openreview.net/attachment?id=4VAF3d5jNg&name
>
> __R-Q3:__
>
> In L99-L101, we have acknowledged that ViTs have been used for UAD in recent works. As discussed in Rebuttal, we consider foundation ViTs as an important and bedstone component of our Dinomaly from a narrative perspective to emphasize the importance of a well-pretrained ViT backbone in UAD context.
>
> According to the "less is more" essence of this paper, we did not intend to propose "novel" or "never seen" technologies (L124-L125), but to propose simple pre-exists elements that have been long ignored for achieving SoTA performance for MUAD.
>
> Most proposed elements (especially noisy MLP, Linear Attention, Loose Constraint) are closely bounded to modern ViTs. Applying such elements to the only previous ViT-based method (ViTAD) would just convert it to Dinomaly. Such elements are extensively evaluated on various ViT variants in Table A4 to show their generalizability. Loose Loss can be directly applied to previous CNN-based methods. Noisy Bottleneck can be adapted to RD4AD with minor modifications (apply dropout before MFF layer). Following your new suggestions, we apply these modules to a different framework to validate the effectiveness of our contributions. The results are shown as follows, where these two elements boost RD4AD to a whole new level.
>
> | Methods | I-AUROC | I-AP | I-F1 | P-AUROC | P-AP | P-F1 | P-AUPRO |
> |---|---|---|---|---|---|---|---|
> | RD4AD | 94.6 | 96.5 | 95.2 | 96.1 | 48.6 | 53.8 | 91.1 |
> | RD4AD+Loose Loss | 98.4 | 99.4 | 97.9 | 97.2 | 58.6 | 60.4 | 92.9 |
> | RD4AD+Noisy Bottleneck | 98.2 | 99.2 | 97.5 | 96.8 | 60.0 | 61.1 | 92.7 |
> | RD4AD+Both | 98.5 | 99.4 | 97.8 | 97.2 | 59.6 | 61.2 | 93.0 |
>
> __R-Q4/6:__
>
> __1)__ As previously discussed, Dinomaly with the resolution of 224x224 has already been presented in Table A4 (last row), in which Dinomaly also achieved SoTA results (I-AUROC=99.3). 256 is not feasible for ViT-Base/14 as 256 is not divisible by 14.  As mentioned in the previous rebuttal, we followed the common comparison strategy based on "optimum vs. optimum" in the manuscript. We will include both settings in the main Table 1.
>
> | Methods | Input Size| I-AUROC | I-AP | I-F1 | P-AUROC | P-AP | P-F1 | P-AUPRO |
> |---|---|---|---|---|---|---|---|---|
> | RD4AD | 256 | 94.6 | 96.5 | 95.2 | 96.1 | 48.6 | 53.8 | 91.1 |
> | ViTAD | 256 | 98.3 | 99.4 | 97.3 | 97.7 | 55.3 | 58.7 | 91.4 |
> | ReContrast | 256 | 98.3 | 99.4 | 97.6 | 97.1 | 60.2 | 61.5 | 93.2 |
> | Dinomaly | 224 | 99.3 | 99.7 | 99.0 | 98.1 | 63.0 | 64.5 | 92.6 |
>
> __2)__ Most prior UAD methods use ConvNets. ConvNets have much smaller downsample stride than ViTs. Therefore, such methods can utilize 64x64 feature maps under 256 input. Empirically, 64x64 feature maps are already saturated for UAD tasks. Further enlarging causes over-focusing on non-semantic low-level noises. (that is why pixel-reconstruction methods are no longer used) On the contrary, ViT/16 can only get 16x16 feature maps given the same input size, which is the reason why ViTAD does not suffer from performance drop when increasing input size. From another perspective, fairness is also questioned given the same image size as CNN-based methods and ViT-based methods have different feature map sizes.
>
> __Questions 3), 4) and 5) are discussed in the next official comments.__

---

> ### Author Response · Authors · 2024-08-11
> **Response (2/2)**
>
> Response (2/2)
>
> __3)__ All metrics are shown as follows. For RD4AD, P-AP and P-F1 increase with resolution at 320, but still do no further benefits from 384; P-AUROC and P-AURPO do not increase with resolution. For ViTAD, P-AP and P-F1 increase with resolution;  P-AUROC and P-AURPO do not increase with resolution. For ReContrast, pixel-level metrics generally do not benefit from increasing resolution. In addition, it is worth considering whether such pixel-level improvement is worth a large decrease in image-level detection performance, as image-level detection is usually more relevant in real applications.
>
> | Methods | Input size | I-AUROC | I-AP | I-F1 | P-AUROC | P-AP | P-F1 | P-AUPRO |
> |---|---|---|---|---|---|---|---|---|
> | RD4AD | 256 | 94.6 | 96.5 | 95.2 | 96.1 | 48.6 | 53.8 | 91.1 |
> | RD4AD | 320 | 93.2 | 96.9 | 95.6 | 95.7 | 55.1 | 57.5 | 91.1 |
> | RD4AD | 384 | 91.9 | 96.2 | 95.0 | 94.9 | 52.1 | 55.3 | 90.8 |
> | ViTAD | 256 | 98.3 | 99.4 | 97.3 | 97.7 | 55.3 | 58.7 | 91.4 |
> | ViTAD | 320 | 98.3 | 99.2 | 97.1 | 97.6 | 61.3 | 63.3 | 92.4 |
> | ViTAD | 384 | 97.8 | 98.9 | 96.3 | 97.5 | 62.5 | 63.7 | 92.4 |
> | ReContrast | 256 | 98.3 | 99.4 | 97.6 | 97.1 | 60.2 | 61.5 | 93.2 |
> | ReContrast | 320 | 98.2 | 99.2 | 97.5 | 96.8 | 61.8 | 62.6 | 93.3 |
> | ReContrast | 384 | 95.2 | 98.0 | 96.4 | 96.5 | 57.7 | 59.5 | 92.6 |
>
>
> __4)__ MAE was also tested as the backbone of ViTAD in their paper, resulting in worse performances (I-AUROC, MAE=95.3 vs. DINO=98.3, Table 1 in ViTAD), which was attributed to the weak semantic expression caused by its pretraining strategy (also note MAE is bad in other unsupervised tasks such as ImageNet kNN). Comparatively, Dinomaly with MAE achieves much better results (I-AUROC=97.3) than ViTAD with MAE.
>
> According to your suggestion, we reproduce ViTAD (original DINO) with the stronger DINOv2 and DINOv2-Register as the backbone. Arming ViTAD with stronger backbones results in slightly better performance, but still lags behind our Dinomaly. It is expected because ViTAD with DINOv2-R is very similar to Dinomaly baseline (Table 3 first row).
>
> | Methods | Backbone | I-AUROC | I-AP | I-F1 | P-AUROC | P-AP | P-F1 | P-AUPRO |
> |---|---|---|---|---|---|---|---|---|
> | ViTAD | MAE | 95.3 | 97.7 | 95.2 | 97.4 | 53.0 | 56.2 | 90.6 |
> | ViTAD | DINO (original) | 98.3 | 99.4 | 97.3 | 97.7 | 55.3 | 58.7 | 91.4 |
> | ViTAD | DINOv2 | 98.7 | 99.4 | 98.1 | 97.6 | 55.3 | 59.1 | 92.7 |
> | ViTAD | DINOv2-R | 98.5 | 99.3 | 97.8 | 97.4 | 54.5 | 59.2 | 92.8 |
>
>
> __5)__ Sorry for the confusion. We did not mean that we do not care about computational cost and building models at all costs. We mean that we did not design modules or techniques aiming to reduce computational cost.
>
> We believe scaling ability is important in this decade, so we armed Dinomaly from ViT-Small to ViT-Large. On the contrary, previous methods did not show this obeying of scaling law (e.g., ViTAD favors ViT-Small over ViT-Base. RD4AD favors ResNet50 over ResNet101). We provide a variety of selections for downstream users to choose from according to their budget.
>
> In Table A2, we have shown that Dinomaly can be scaled down to ViT-Small with the computation cost (26.3G) given 396x396 input, which is already lower than RD4AD given 256x256 input (28.4G). This computation cost is very promising for real industrial applications, yielding latency of 6.8ms per image (~147 FPS) on home-use-level NVIDIA 3090.
>
> Moreover, we further scale down Dinomaly-ViT-Small by decreasing input size, as shown below, where Dinomaly produces SoTA results given more limited computation budgets.
>
>
> | Methods | Parameters | MACs | I-AUROC | I-AP | I-F1 | P-AUROC | P-AP | P-F1 | P-AUPRO |
> |---|---|---|---|---|---|---|---|---|---|
> | RD4AD | 80.6M | 28.4G | 94.6 | 96.5 | 95.2 | 96.1 | 48.6 | 53.8 | 91.1 |
> | ViTAD | 39.0M | 9.7G | 98.3 | 99.4 | 97.3 | 97.7 | 55.3 | 58.7 | 91.4 |
> | ReContrast | 154.2M | 67.4G | 98.3 | 99.4 | 97.6 | 97.1 | 60.2 | 61.5 | 93.2 |
> | Dinomaly-ViT-Small-396x396 | 37.4M | 26.2G | 99.3 | 99.7 | 98.7 | 98.1 | 68.3 | 67.8 | 94.4 |
> | Dinomaly-ViT-Small-280x280 | 37.4M | 14.5G | 99.3 | 99.7 | 98.7 | 98.0 | 65.1 | 65.7 | 93.4 |
> | Dinomaly-ViT-Small-252x252 | 37.4M | 11.6G | 99.1 | 99.6 | 98.7 | 98.0 | 63.9 | 64.9 | 93.1 |
>
>
> We believe scaling ability is important in this decade. By setting the current upper-bounds on various benchmarks, we aim to (and can) provide a scalable framework that can accommodate different demands of application needs. If someone's main concern is detection performance and has abundant computation resources, they are given the option to choose Dinomaly-ViT-Large/Base for ultimate performance. If someone has limited computation resources or cares about FPS/latency, they can choose Dinomaly-ViT-Small while still producing state-of-the-art performance.
>
> Again, thank you for your thorough comments and review. Looking forward to further discussion with you.

---

> > ### Comment · Reviewer_LXBB · 2024-08-12
> > **For Authors**
> >
> > Thank you for the detailed response. I suggest that the author add the fairness resolution experiments to the revised version and include results from the VisA and Real-IAD datasets. I have also reproduced the Dinomaly-ViT-Small experimental results using the official code. The author could consider uploading the logs of the main experiments (excluding ablation studies) to the website to facilitate replication and comparison by future researchers.
> >
> > Although I still maintain a somewhat negative view of the paper's technological contributions, the solid experimental results could positively impact the community. Therefore, I have decided to raise my score, but the author should adhere to their commitment to address the issues in the revised version.

---

> > > ### Author Response · Authors · 2024-08-14
> > > **Response**
> > >
> > > Sincere thanks for your reply. We will do so. Again, thank you for your thorough review and valuable discussion.

---

### Official Review · Reviewer_xmFt · 2024-07-16

**Soundness:** 3
**Presentation:** 3
**Contribution:** 3
**Rating:** 5
**Confidence:** 3

**Summary:**

This paper introduces Dinomaly, a minimalistic unsupervised anomaly detection (UAD) method designed to bridge the performance gap between multi-class UAD and class-separated UAD. Utilizing pure Transformer architectures with key components such as Foundation Transformers, Noisy Bottleneck, Linear Attention, and Loose Reconstruction, Dinomaly achieves superior performance on MVTec-AD, VisA, and Real-IAD benchmarks, surpassing state-of-the-art methods.

**Strengths:**

1. Dinomaly effectively bridges the performance gap between multi-class and class-separated UAD, achieving superior results on popular benchmarks such as MVTec-AD, VisA, and Real-IAD.
2.  It utilizes a simple, straightforward approach with pure Transformer architectures, avoiding complex modules or specialized tricks.
3. The detailed ablation study demonstrates the effectiveness of each component—Noisy Bottleneck, Linear Attention, Loose Constraint, and Loose Loss—in enhancing anomaly detection.

**Weaknesses:**

1. The method might be perceived as too application-oriented, lacking broader theoretical contributions.

**Questions:**

See the weaknesses.

**Limitations:**

See the weaknesses.

---

> ### Author Rebuttal · Authors · 2024-08-02
>
> Thank you for your valuable reviews and comments.
>
> We appreciate the concern about the balance between application and theory in our work. While Dinomaly does focus on practical applications and SOTA results, we believe it makes theoretical contributions to the field of unsupervised anomaly detection, providing conceptual insights on what is "identical mapping" and how to mitigate this phenomenon that serves as the chief obstacle of  UAD tasks.
>
> For the first time, we attribute the "identical mapping" phenomenon to the over-generalization nature of neural networks. Accordingly, we discovered multiple key components and operations that can both theoretically and empirically alleviate over-generalization in UAD context. In addition, we challenge the conventional views: more complex architectures are necessary for better performance in anomaly detection tasks.

---

### Official Review · Reviewer_cd2L · 2024-07-16

**Soundness:** 4
**Presentation:** 3
**Contribution:** 3
**Rating:** 6
**Confidence:** 3

**Summary:**

Dinomaly simplifies the anomaly detection process by eliminating the need for complex designs, additional modules, or specialized techniques. It relies solely on basic Transformer components such as self attention mechanisms and multi-layer perceptrons (MLPs) to perform anomaly detection for multi class images.

**Strengths:**

This paper has a clear motivation and contribution. The paper effectively proposes the viewpoint of "less is more" in multi class unsupervised anomaly detection, emphasizing how the simplicity of model architecture can achieve or surpass the performance of more complex systems.

**Weaknesses:**

I hope to provide a specific explanation of the information provided by the decision-making process for identifying anomalies in the model.

**Questions:**

Is the viewpoint presented in this article too sharp? Large scale models may not necessarily lack their advantages. I acknowledge this work and would like to see other people's questions and the author's response to determine the final score.

**Limitations:**

The author candidly acknowledged the limitations of the work and provided the problems that need to be addressed.

---

> ### Author Rebuttal · Authors · 2024-08-02
>
> Thank you for your valuable reviews and comments.
>
> __W1: Decision-making to identify anomalies.__
>
> Dinomaly is based on the assumption that the networks respond differently during inference between seen and unseen input, faithfully reconstructing normal regions while failing for anomalous regions. We depict the calculation procedure of anomaly activation maps in the PDF of the rebuttal. We believe this figure can clearly demonstrate how the model identifies anomaly regions. It will be included in the paper in the final version.
>
> __Q1: Minimalism and large-scale models.__
>
> The "less is more" viewpoint of this work does not mean "small-scale model" (parameters and computation). The model scale of Dinomaly is actually pretty large compared to existing UAD methods, which can be further scaled up to ViT-Large (Table A2) obeying the "scaling law". On the contrary, the "less is more" philosophy is embodied in the minimalistic design of architectures and tricks, demonstrating the power of plain and simple (not small) framework in UAD tasks.

---

### Author Rebuttal · Authors · 2024-08-06

First, thank all reviewers for their valuable reviews and comments.

Please post any new questions in the 7-day discussion period.

---

### Decision · Program_Chairs · 2024-09-25

**Decision:**

Reject

**Comment:**

This paper presents a minimalistic reconstruction-based anomaly detection framework for multi-class unsupervised anomaly detection. The framework comprises four key components: foundation transformers, noisy bottleneck, linear attention, and loose reconstruction. The proposed method has been extensively tested on MVTec-AD, VisA, and Real-IAD datasets, demonstrating state-of-the-art performance in multi-class unsupervised anomaly detection scenarios.

The paper receives mixed scores, including three positive and two borderline. All reviewers concur that the paper introduces a simple and effective method for multi-class anomaly detection, and achieves state-of-the-art performance (i.e., anomaly detection accuracy) on multiple datasets. The paper is well-written, well-structured, and easy to understand. However, there are contentious points primarily concerning the novelty of the proposed method, the fairness of the comparison, and the omission of a discussion on the number of parameters, computational complexity, or time complexity.

**Limited Novelty**. The proposed method uses the existing RD4AD framework with several new modules including foundation transformers, noisy bottleneck, linear attention, group loss, and loose loss. Notably, the noisy bottleneck is an MLP with dropout. And it is, in fact, very similar to some existing reconstruction works that randomly mask input pixels or features. The difference lies in the proposed method performing random dropout in the channel dimension while existing methods perform random mask in the spatial dimension. Furthermore, linear attention is an existing work, while group loss and loose loss are simple improvements to the existing reconstruction loss.

**Unfair Comparisions**. Compared to existing methods (e.g., RD4AD and UniAD), this paper employs a more powerful pre-trained model (DINO-v2 vs WideResNet50 or EfficientNet-b4) and a larger input resolution (448$\times$448 vs. 224$\times$224 or 256$\times$256). Consequently, it is currently hard to directly evaluate the actual performance improvement due to the unfair comparison. I suggest that the authors first establish strong baselines using the same backbone and input resolution, then compare the proposed method and the state-of-the-art. Additionally, using a larger backbone and higher input resolution implies more hidden computational costs, such as the number of parameters, memory, and inference time. However, the detailed comparisons are absent.

Although the author has partially addressed the above contentious points in the rebuttal period, I believe it will be challenging to fully resolve these issues in the camera-ready version. Therefore, I lean towards rejecting it and encourage the author to fully consider the reviewer's suggestions and revise the paper for submission to other venues.